# Validation of the UNESP-Botucatu pig composite acute pain scale (UPAPS)

**Stelio Pacca Loureiro Luna** [1☯*], **Ana Lucélia de Araújo** [2☯¤a], **Pedro Isidro da Nóbrega Neto** [3‡], **Juliana Tabarelli Brondani** [1¤b‡], **Flávia Augusta de Oliveira** [2¤c‡], **Liliane Marinho dos Santos Azerêdo** [3‡], **Felipe Garcia Telles** [1‡], **Pedro Henrique Esteves Trindade** [1‡]

**1** Department of Veterinary Surgery and Animal Reproduction, School of Veterinary Medicine and Animal Science, São Paulo State University (Unesp), Botucatu, São Paulo, Brazil, **2** Post graduation Program in Anaesthesiology, Medical School, São Paulo State University (Unesp), Botucatu, São Paulo, Brazil, **3** Health and Rural Technology Centre, Federal University of Campina Grande (UFCG), Patos, Paraíba, Brazil

☯ These authors contributed equally to this work.
¤a Current address: Federal Institute of Education Science and Technology of Paraíba (IFPB), Veterinary Medicine School, Sousa, Paraíba, Brazil
¤b Current address: Clínica Veterinária Santa Quitéria, Curitiba, Paraná, Brazil
¤c Current address: School of Veterinary Medicine and Animal Science, Federal University of Tocantins, Araguaína, Tocantins, Brazil
‡ These authors also contributed equally to this work.
* stelio.pacca@unesp.br

**Data Availability Statement:** All relevant data are within the manuscript and its Supporting Information files.

## Abstract

The creation of species-specific valid tools for pain assessment is essential to recognize pain and determine the requirement and efficacy of analgesic treatments. This study aimed to assess behaviour and investigate the validity and reliability of an acute pain scale in pigs undergoing orchiectomy. Forty-five pigs aged 38±3 days were castrated under local anaesthesia. Behaviour was video-recorded 30 minutes before and intermittently up to 24 hours after castration. Edited footage (before surgery, after surgery before and after rescue analgesia, and 24 hours postoperatively) was analysed twice (one month apart) by one observer who was present during video-recording (in-person researcher) and three blinded observers. Statistical analysis was performed using R software and differences were considered significant when $p<0.05$. Intra and inter-observer agreement, based on intra-class correlation coefficient, was good or very good between most observers (>0.60), except between observers 1 and 3 (moderate agreement 0.57). The scale was unidimensional according to principal component analysis. The scale showed acceptable item-total Spearman correlation, excellent predictive and concurrent criterion validity (Spearman correlation $\geq 0.85$ between the proposed scale *versus* visual analogue, numerical rating, and simple descriptive scales), internal consistency (Cronbach's $\alpha$ coefficient >0.80 for all items), responsiveness (the pain scores of all items of the scale increased after castration and decreased after intervention analgesia according to Friedman test), and specificity (> 95%). Sensitivity was good or excellent for most of the items. The optimal cut-off point for rescue analgesia was $\geq$ 6 of 18. Discriminatory ability was excellent for all observers according to the area under the curve (>0.95). The proposed scale is a reliable and valid instrument and may be used

**Funding:** SPLL - Financial support from the Foundation for Research Support of the State of São Paulo – FAPESP (Process 2010/08967-0 and 2017/12815-0). http://www.fapesp.br/en/ ALA - Grant from REUNI -http://reuni.mec.gov.br/ The funders had no role in study design, data collection and analysis, decision to publish, or preparation of the manuscript.

**Competing interests:** The authors have declared that no competing interests exist.

clinically and experimentally to assess postoperative acute pain in pigs. The well-defined cut-off point supports the evaluator's decision to provide or not analgesia.

## Introduction

Pigs have two main utility purposes for humankind: meat source and translational medicine. With regard to livestock, pork is by far the most consumed meat globally, 25% and 100% above chicken and beef respectively [1]. However, among production animals, pigs are the most neglected with regard to pain assessment and treatment. While approximately 80% of Canadian veterinarians considered their knowledge sufficient to assess pain in large animals [2], only 32% of pig veterinarians considered their knowledge adequate in this area [2]. Approximately 40% of UK veterinarians working with swine consider that "it is difficult to recognise pain in pigs" [3]. This probably justifies the fact that among large animals, the swine species receive considerably less analgesics than bovine and equine species [2]. For example, in Canada, after orchiectomy less than 0.001% of piglets receive analgesia, compared with 7% of beef and 19% of dairy calves younger than 6 months of age [2].

Orchiectomy is one of the most frequent surgical procedures in pigs, however the majority of piglets are castrated without anaesthesia [4,5]. Although this practice promotes physiological and behavioural changes strongly suggestive of acute pain [6–8], the main argument to sustain this animal-welfare compromising practice is the small profit margin of the industry [2,3]. This paradigm has been challenged by the fact that pigs treated with anaesthesia and analgesia showed increased short [9] and long-term weight gain [10], when compared to pigs castrated without anaesthesia, thus, providing an economic benefit to producers [10].

Translational medicine is the other side of the coin for using pigs. Swine have progressively replaced the use of dogs and non-human primates as non-rodent models for research and teaching, due to several limitations related to using other species, including ethical issues [11,12]. The number of pigs used for research in Canada in 2018 was 32,231, against 14,283 dogs and 8,522 non-human primates [13]. These numbers were 6,569 in USA in 2017 [14], and 72,749 in Europe in 2014 [15]. The similar anatomophysiological characteristics of swine compared to man, make them promising animal models for several purposes [11,12,16], including xenotransplantation [11,12], and accurate models for pain studies [17], including neuropathic pain [18].

The scenario of the use of analgesics in pigs used for research is not that different from that used in livestock. As animal models for humans, pigs experience a similar degree of pain under experimental painful conditions; therefore pain should be mitigated based on ethical grounds. A recent review about the use of analgesia in pigs submitted to experimental surgery showed that from 233 studies that met the inclusion criteria, information on postoperative analgesia and pain assessment was reported in only 37% and 10% of them respectively [19], showing that pain measurement and treatment were either neglected or not reported. This finding suggests the great necessity for the development of pain assessment tools to fill this gap and guarantee the welfare of pigs undergoing research. Recognition and assessment of pain, followed by analgesic treatment, guarantees one of the three "Rs", i.e., refinement, in pigs submitted to painful procedures [19].

The lack of a validated instrument to assess pain in pigs may account for the restricted use of analgesics in these species in both production and research. The development of species-specific validated pain assessment instruments is essential to recognize and quantify pain and to determine the requirement and efficacy of analgesic treatment in animals [17].

Pain or nociception may be assessed through objective or subjective methods [20]. The advantages of subjective methods based on pain-related behaviour analysis are that it is not invasive, does not require equipment or restraint, and may be assessed by remote observation. These features suggest that pain-related behaviour is the principal and easiest tool to evaluate pain in animals.

The validation process should encompass the investigation of species-specific behaviour [20–21,22] followed by assessment of intra and inter-rater reliability, content validity, which assesses if each item is representative, responsiveness or construct validity, internal consistency, item-total correlation, criterion validity, specificity, sensitivity, and definition of a cut-off point for analgesic intervention [20–23].

Although the authors recognize the great effort paid in assessing and describing pain-related behaviours in pigs, to our knowledge, to date, there is no validated behavioural instrument developed to assess pain in these species, like those validated in the most common domestic species, such as dogs [24,25], cats [21,26], cattle [22], and horses [27–29]. Recent studies have assessed facial expression to develop a Grimace scale for pigs [30,31], however to our knowledge neither a facial or behavioural pain scale have yet been fully validated. Therefore, this study aimed to develop and validate a scale to assess postoperative acute pain in pigs undergoing castration based on behavioural analysis.

## Methods

The study was approved by the Ethical Committee for the Use of Animals in Research of the School of Veterinary Medicine and Animal Science, Unesp, Botucatu, Brazil, under protocol number 102/2014 and follows the Brazilian Federal legislation of CONCEA (National Council for the Control of Animal Experimentation). This is a prospective, randomized, and blind study conducted at the University farms of the authors´ institutions.

Inclusion criteria were based on clinical examination and laboratorial assessment (haemogram and serum biochemistry—plasma urea, creatinine, alkaline phosphatase, and glutamic pyruvic transaminase) to guarantee they were healthy. Pigs that were not clinically healthy or did not present normal laboratorial data were excluded. Pigs that presented any postoperative complications, clinical problems, such as diarrhoea, or were bruised due to fights during the 24 hours of assessment were excluded.

Pigs were selected from the University commercial production. A pilot study was carried out with 12 piglets aged 35 days, submitted to castration only to define the experimental protocol and to recognize pain-related behaviours. For the main study, 45 Landrace, Large White, Duroc, and Hampshire male pigs were randomly selected. The pigs were aged 38±3 days (range 35–41 days) and weighed 11.06 ± 2.28 kg. Groups of five pigs were allocated in suspended collective iron pens 15 days prior to the surgical procedure for adaptation to the personnel and environment. The pens measured 2.40 x 1.50 x 1.50 meters (length x width x height), and were located side by side, separated by bars. The piglets received commercial feed three times a day and water was available *ad libitum* in nipple drinkers.

Familiarity with the in-person researcher started during the maternity period when piglets were 14 days old. After a week of contact with the piglets, by cleaning the pen and providing food, the in-person researcher initiated direct friendly contact inside the stall, for 15 minutes three times a day, without making any sudden movement, or using vocal communication, so that the piglets could spontaneously approach.

After the adaptation period, procedures were performed between 8 and 10 AM. The pigs were physically restrained and after local antisepsis, they were submitted to bilateral local anaesthesia with 0.5 mL of 1% lidocaine without vasoconstrictor (Xylestesin®, Cristália,

Itapira, São Paulo, Brazil) injected subcutaneously at each incision line, parallel to the scrotum shaft, followed by 1 mL injected intratesticularly at each testicle. After five minutes, orchiectomy was performed by the same surgeon. Four hours after the surgical procedure, the in-person researcher injected analgesic rescue with 2 mg/kg of flunixin meglumine (Flunixin®, Chemitec, São Paulo, Brazil) and 0.5 mg/kg of morphine (Dimorf®, Cristália, Itapira, São Paulo, Brazil) intramuscularly (IM) in the cervical region. At the end of the observation period of 24 hours, the animals received 2 mg/kg of flunixin meglumine IM, once a day for two consecutive days. The in-person researcher treated the surgical wounds topically with silver sulfadiazine (Bactrovet®, König, São Paulo, São Paulo, Brazil), once a day for five days and assessed the surgical wound every 48 hours thereafter, for 10 days.

The pigs were evaluated from 24 hours to 16 hours before surgery and at 2, 4, 6, 8, and 24 hours after castration. Cameras were positioned on the ceiling of the pens to record 30-minute recordings at each of these moments, for a total of 135 hours. The pigs had no visual contact with the in-person researcher. After video recording, if the food was not available at the feeder, the researcher provided a small amount of food, and a short period of video was recorded to evaluate appetite. After the end of the field study pigs were kept until 145 days of age when they were submitted to humane slaughter accomplished by electrical stunning, followed by exsanguination, according to the Brazilian Federal legislation (Ministry of Agriculture).

The videos from the main study were observed for the first time for recognition of different behaviours and elaboration of an ethogram, based on previous studies [6,8,32–34]. After this first analysis, four moments were selected as the most representative as described in Fig 1. The footage corresponding to each of these moments was observed again to measure the duration and frequency of each pain-related behaviour.

The scale was developed according to the analysis of the relevance of pain-related behaviours previously described in the literature [6,8,32–35] and behaviours observed both in the pilot study and during video edition by the in-person researcher.

| MOMENTS | M1 | | M2 | | M3 | M4 | |
|---|---|---|---|---|---|---|---|
| **Video recordings** | From 24 to 16 h before surgery | Surgery | 3.5 to 4h after surgery (before rescue analgesia) | Rescue analgesia Morphine and flunixin IM | 5.5 to 6 h after surgery (1,5 to 2 h after rescue analgesia) | 24 h after surgery | Flunixin IM |
| **Expected pain condition** | Free of pain | | Intense | | Mild (pain alleviated by analgesia) | Moderate (analgesia effect was fading) | |

**Fig 1. Timeline of interventions, drugs and pain assessments performed for validation of the Unesp-Botucatu pig composite acute pain scale (UPAPS).**

Once the initial proposed scale had been defined, content validity was assessed by three evaluators experienced in assessing pain in other domestic species, who scored each behaviour according to the degree of importance related to pain (-1 = irrelevant item; 0 = do not know; 1 = relevant item). Items which reached an average score $\geq$ 0.5 were included in the scale [21,22,36] From this point the scale was considered ready for video analysis. The proposed scale was composed of six behavioural items. Each item presented four descriptive levels. A numerical score was attributed from 0 to 3, where "0" reflected normal animal state (free of pain) and "3", the maximum value, corresponded to accentuated behavioural alteration. The maximum score of the entire scale was 18 points.

The in-person researcher edited each of the four 30-minute videos into 4-minute videos representing the 4 moments previously described (Fig 1). The 180 edited videos were analysed by four observers, two men (SPLL and PINN) and two women (FAO and ALA). The first two are the senior authors, with above 20-year experience in anaesthesia and pain assessment in farm animals. FAO developed the cattle postoperative pain scale [22] and ALA was considered the "gold-standard" as she was the in-person researcher responsible for video recording and editing as well as creation of the ethogram. All observers watched the videos in a random order, without knowledge about the moment they were observing (blind analysis).

After watching each video, the observers, based on their clinical experience, initially stated if they would provide rescue analgesia at that moment or not. Subsequently, they were required to ascribe a score for the visual analogue scale (VAS), numerical rating scale (NRS; 0 "no pain" to 10 "worst possible pain"), the simple descriptive scale (SDS; 0—no pain to 3— intense pain) and UPAPS. After one month, each observer blindly analysed the footage again with a new order of moments and pigs, to establish the intra-observer reliability.

## Statistical analysis

For the ethogram analysis the time spent in minutes and the percentage of each behaviour before and after surgery were compared by the Friedman test. Differences were considered significant when p < 0.05.

Validation of the pain scale was processed as described before [21,37]. The intra- and inter-observer reliability were assessed by comparing the data between the first and second analysis for each observer and by comparing the degree of agreement among different observers respectively. To define, for the sum of the scores, the inter-and intra-observer reliability, the intraclass correlation coefficient (ICC) was used. ICC = 1 indicates high reliability (no error), whereas ICC = 0 indicates no reliability. The 95% confidence interval (CI) was calculated for each ICC value with 95% CI [22] at all moments assembled. The weighted kappa coefficient was used to calculate the correlation of each item of the scale among the observers, encompassing all moments of assessment. The values obtained were interpreted by Altman's classification as: 0.81–1.0 very good; 0.61–0.8 good; 0.41–0.6 moderate; 0.21–0.4 reasonable; and <0.2 poor [38].

The criterion validity was analysed by concurrent and predictive validity. Concurrent validity was investigated by comparing the scores obtained by the proposed scale against those determined by VAS, NRS, and SDS [21,22,39]. The interpretation of the Spearman's correlation coefficient was defined by the calculation of the results of each observer, as well as for all observers together. A second method to measure concurrent criterion validity was by assessing the agreement between the gold-standard assessor and the other observers. The weighted kappa coefficient was calculated with a CI of 95% [40] for each item of the scale, encompassing all assessment moments. For the total score of UPAPS, the intraclass correlation coefficient (ICC) type "consistency" was used, and its 95% CI. The results of the kappa coefficient and ICC were interpreted [37] according to Altman's classification [38].

Predictive criterion validity was evaluated by the number of pigs that should receive rescue analgesia consistent with the Youden Index (described below) at first moment after surgery when pigs should express the most intense pain (M2). Fisher´s exact test was used to compare the indications of rescue analgesia based on the Youden index *vs* clinical experience.

Principal component analysis defined the number of factors (dimensions or domains) determined by different variables and establish the extension of the scale [37,41,42]. The appreciation of the main components subsidized the execution of the principal component analysis and the factors were based on Kaiser's criterion, which indicates maintenance of all components with eigenvalues > 1 [43]. The factorial structure was confirmed when items showed a factor load $\geq$ 0.50 or $\leq$ -0.50.

Item-total correlation based on the Spearman coefficient was performed to assess homogeneity and relevance of each item of the scale. Each item was correlated with the sum of all scale items, excluding that item, to avoid inflating the results. Values between 0.3 and 0.7 were accepted [23].

The internal consistency of the scale, to determine the interrelation among the items in the instrument, was assessed by calculation of Cronbach's alpha coefficient [44]. Values were considered as follows: 0.60–0.64, minimally acceptable; 0.65–0.69, acceptable; 0.70–0.74, good; 0.75–0.80, very good; and > 0.80, excellent [45].

Construct validity was determined by the hypothesis test methodology. The first hypothesis is that if the scale truly measures pain, the scoring after surgery should be higher than the preoperative score (M1 *versus* M2). The second and third hypotheses are that the score should decrease after analgesia and over time (M2 *versus* M3 and M2 *versus* M4, respectively). The values were expressed in medians and significance was analysed by the Friedman's test [21,22]. According to this analysis, it was possible to assess the response capacity (responsiveness) of the scale.

Specificity was assessed at M1, considering that the piglets were free of pain and reflected the true negative results. The scores at M1 were transformed into dichotomous items and entered into the equation: Specificity = TN/TN+FP, where TN = number of true negatives (score 0, indicating that piglets were not expressing pain); FP = false positives (scores 1, 2, or 3, indicating that piglets were expressing pain before surgery, when they should be supposedly pain-free). Sensitivity was calculated at M2, considering that this was the time pain would be the greatest and reflect the true positive results. Likewise for specificity, scores at M2 were transformed into dichotomous variables and the following equation applied: Sensitivity = TP/TP+FN, where TP = true positives (scores 1, 2, or 3, indicating that piglets were expressing pain after surgery, as expected), FN = false negatives (score 0, indicating that piglets were not expressing pain, when they should be expressing pain after surgery). Specificity and sensitivity were considered excellent when 95–100%; good when 85–94.9%; moderate when 70–84.9%; and not-specific or not-sensitive when <70% [29].

The frequency distribution of the presence of scores 0, 1, 2, and 3 of each item at each moment was assessed by descriptive statistical analysis according to the 2nd phase of the gold standard video analysis.

The data relative to the indication of rescue analgesia were used to determine the optimal cut-off point, that is, the minimum score suggestive of the need for analgesic rescue or intervention. This determination was based on the Youden index, which determines the highest sensitivity and specificity value concurrently from the Receiver Operating Characteristic (ROC) curve [21,22], providing a graphic image of the relation between the "true positives" (sensitivity) and the "false positives" (specificity). The discriminatory capacity of the test was determined by the area under the curve (AUC) [46,47]. AUC values above 0.9 represent high precision. In addition, the diagnostic uncertainty zone was determined by two methods

[48,49]: 1) calculating the 95% confidence interval (CI) by replicating the original ROC curve 1000 times according to the bootstrap method and 2) calculating the sensitivity and specificity value > 0 90. The lowest and highest values of these two methods among all evaluators was assumed to be the diagnostic uncertainty zone [50].

For determination of pain, intensity scores of the 2nd phase from all observers were classified as no pain, mild, moderate or intense pain, at the time of most intense pain (M2). Non-hierarchical cluster analysis was performed, applying the "maximum" distance and the "Ward. D2" method using the total score of UPAPS and NS [51], followed by the Kruskal-Wallis test to assess the difference between the groups.

Statistical analysis was performed using R software in the Rstudio integrated development environment (Version 1.0.143—©2009–2016, Rstudio, Inc.). Differences were considered significant when $p < 0.05$.

## Results

Seventy-eight pigs were castrated, but only data from 45 pigs were processed. Six pigs were excluded due to problems in capturing the footage images and the others due to the previously mentioned exclusion criteria. The surgeries and 135 hours of video recordings were carried out with success. Footage from 40 out of 45 pigs was used for the ethogram because in 5 pigs footages were recorded for slightly less than 30 minutes at some moments; therefore data from these 5 five pigs were excluded from the ethogram analysis. For validation of the scale, 180 videos of approximately 4-minute durations were edited and footage of the main study from 45 pigs were used. There were no postoperative complications in these pigs.

### Behaviour data

The behaviours observed in the ethogram are described in Table 1.

Where normal behaviours are concerned, piglets spend a longer time eating and drinking water after rescue analgesia, compared to before rescue analgesia (Table 2). When compared to values before surgery, piglets spent a shorter time walking, resting, sleeping, rooting, and interacting and a longer time in abnormal posture, walking with difficulty, and lying alone or with discomfort after surgery. Rescue analgesia reverted the duration of these behaviours to values observed before surgery, except for resting and sleeping, which remained lower after rescue analgesia compared to before surgery.

### Pain scale data

The scale constructed for video analysis was composed of six items, each with four sub-items, providing a maximum score of 18 points (Table 3).

### Intra-observer reliability

The intra-observer reliability, including all assessed moments, was good for all evaluators, except for attention to the affected area and miscellaneous behaviours which was moderate for evaluator 3 (Table 4).

### Inter-observer reliability

Reproducibility based on matrix correlation ranged from moderate (Observer 1 *vs.* 3) to very good (Observer 1 *vs.* Gold standard) (Table 5).

**Table 1. Description of the behaviours evaluated in the ethogram of pigs submitted to orchiectomy (adapted from the literature [6, 8, 32, 33, 34]).**

| Behaviour | Description |
|---|---|
| **Normal** | |
| Stand inactive | Standing still, without expression of other normal behaviours |
| Walk | Walking |
| Sit | Seated. |
| Lie down | At rest, lying down |
| Sleep | Sleeping |
| Eat | Eating from the trough |
| Drink | Drinking water |
| Root | Exploring the environment, rooting around the pen, or lying down |
| Urinate/defecate | Act of urinating or defecating |
| **Friendly social interactions** | |
| Lick | Perform small chewing movements while touching another animal |
| Smell | Smelling another piglet standing or lying down |
| **Aggressive social interaction** | |
| Fight | Direct physical confrontation, with head butts, pushes, bites and chases, in rapid and continuous succession |
| **Stereotypical behaviour** | |
| Bite | Biting parts of the pen (bars, trough) |
| Run/Agitated | Visibly stressed, running around the pen; disturbed, with escapist behaviour |
| **Behaviour indicative of pain** | |
| Altered posture | Presenting one or more repetitions of abnormal behaviours: standing kicking, scratching/rubbing the affected area (surgical wound), protecting the affected area (surgical wound), contracted muscles, arched back, and energetic tail wagging |
| Difficulty walking | Difficulty walking, apparent discomfort in the perineal region, contraction of scrotal region, and energetic tail wagging |
| Lie down alone | Quiet, lying alone in the pen, not allowing interaction |
| Lie down uncomfortably | Changes posture with apparent discomfort, tail wagging and/or rubbing the affected area (surgical wound) on the floor |

## Criterion validity

**Concurrent criterion validity based on the agreement between the gold-standard evaluator and the other observers.** The criterion validity based on the agreement between the gold-standard evaluator and the three other observers was good or very good for the total score of the UPAPS and the kappa correlation coefficient ranged from reasonable to good for each item of the scale (Table 6).

**Concurrent criterion validity based on the correlation between the UPAPS and unidimensional scales.** High correlations were observed between the scores of the proposed scale and the VAS (r = 0.846), NRS (r = 0.878), and SDS (0.854) (Table 7) for all evaluators.

**Predictive criterion validity.** Indication of rescue analgesia was higher when based on the Youden index compared to clinical experience (Table 8). According to the Youden index, all the grouped evaluators would indicate rescue analgesia in 84% (71–98%) of the pigs in M2 (Table 8), thus confirming predictive criterion validity. Otherwise, needless analgesia would be indicated in 6% (0–11%) of the pigs at M1. Therefore, UPAPS demonstrated sensitivity to diagnose pain and specificity to differentiate pigs not suffering pain.

**Table 2. Median and amplitude of the number of minutes/30 minutes of normal behaviours, friendly social interactions, aggressive social interactions, stereotypical behaviours, and pain-related behaviours in piglets submitted to castration (n = 40).**

| Items of the Ethogram | M1 (%) | M2 (%) | M3 (%) | M4 (%) |
|---|---|---|---|---|
| Eat | 23$^{ab}$ (0–91) | 21$^b$ (0–51) | 30$^a$ (7–93) | 24$^{ab}$ 1–68) |
| Drink | 2$^b$ (0–9) | 1$^b$ (0–9) | 4$^a$ (0–20) | 2$^{ab}$ (0–12) |
| Stand | 3$^a$ (0–18) | 1$^{ab}$ (0–23) | 0$^b$ (0–10) | 1$^{ab}$ (0–8) |
| Walk | 7$^a$ (0–38) | 2$^b$ (0–19) | 8$^a$ (0–50) | 3$^{ab}$ (0–23) |
| Lie down at rest | 22$^a$ (3–79) | 9$^b$ (0–41) | 5$^b$ (0–68) | 13$^b$ (0–56) |
| Sleep | 0$^a$ 0–70) | 0$^b$ (0–11) | 0$^b$ (0–23) | 0$^{ab}$ (0–50) |
| Sit | 0 (0–1) | 0 (0–1) | 0 (0–1) | 0 (0–10) |
| Root | 7$^{ab}$ (0–23) | 1$^c$ (0–27) | 9$^a$ (0–40) | 4$^{bc}$ (0–19) |
| Urinate/defecate | 0 (0–3) | 0 (0–4) | 0 (0–3) | 0 (0–2) |
| Interact | 5$^a$ (0–40) | 0$^c$ (0–10) | 3$^{ab}$ (0–41) | 0$^{bc}$ (0–20) |
| Fight | 0 (0–6) | 0 (0–7) | 0 (0–17) | 0 (0–12) |
| Bite the bars or objects | 0 (0–6) | 0 (0–7) | 0 (0–60) | 0 (0–17) |
| Stand: altered posture | 0$^c$ (0–0) | 6$^a$ (0–20) | 0$^b$ (0–20) | 6$^a$ (0–24) |
| Altered walk | 0$^c$ (0–0) | 4$^a$ (0–17) | 0$^b$ (0–14) | 2$^a$ (0–32) |
| Run/agitated | 0 (0–0) | 0 (0–11) | 0 (0–27) | 0 (0–16) |
| Lie down alone | 0$^c$ (0–4) | 16$^a$ (0–78) | 0$^{bc}$ (0–49) | 0$^b$ (0–59) |
| Lie down uncomfortably | 0$^d$ (0–0) | 20$^a$ (0–70) | 0$^c$ (0–24) | 8$^b$ (0–57) |

M1: preoperative; M2: postoperative, before rescue analgesia; M3: postoperative, after rescue analgesia; M4: 24 hours postoperative.

$^{a,b,c,d}$ different superscript letters indicate statistical difference between time periods, where a>b>c>d.

## Principal component analysis

The principal component analysis produced a factor with a value of 3.47. All items presented an acceptable factor load (Table 9, Fig 2).

## Correlation coefficient of item score with total score (item-total score)

The item-total correlation for all moments varied from 0.42 to 0.87 (Table 10). All items were accepted as they were above 0.3 (23).

## Internal consistency

The Cronbach's $\alpha$ coefficient above 0.80 for all items indicated excellent internal consistency and reinforced the possibility of using the total score to interpret the results (Table 11).

## Construct validity (responsiveness)

When compared to the basal values (M1), except for observer 2, pain scores increased significantly after surgery (M2), decreased after analgesia (M3), and were intermediate (greater than M1 and M3, but smaller than M2) 24 hours after the surgery (M4) (Table 12). Therefore, the scale shows excellent responsiveness, as it was even possible to differentiate intense (M2) from moderate pain (M4).

When each item was considered separately, gold standard observer scores increased significantly after surgery and decreased after rescue analgesia, showing that all items presented responsiveness and, therefore, construct validity (S1 Table). All sub-items of attention to

**Table 3. The UNESP-Botucatu composite pain scale for assessing postoperative pain in pigs.**

| Item | Score/criterion | Links to videos |
|---|---|---|
| **Posture** | (0) normal (any position, apparent comfort, relaxed muscles) | https://youtu.be/-loODGUwmS0 |
| | (1) changes posture, with discomfort | https://youtu.be/V5zEiuJnF_g |
| | (2) changes posture, with discomfort, and protects the affected area | https://youtu.be/PAZplCKxuhk |
| | (3) quiet, tense, and back arched | https://youtu.be/qubgsQeoQ-8 |
| **Interaction and interest in the surroundings** | (0) interacts with other animals; interested in the surroundings | https://youtu.be/OwbrMRogO-I |
| | (1) only interacts if stimulated by other animals; interested in the surroundings. | https://youtu.be/IPdIeVaeDlY |
| | (2) occasionally moves away from the other animals, but accepts approaches; shows little interest in the surroundings | https://youtu.be/kQ0gZ4CF5Zk |
| | (3) moves or runs away from other animals and does not allow approaches; disinterested in the surroundings | https://youtu.be/ZrHZZlLk7Q4 |
| **Activity** | (0) moves normally | https://youtu.be/3_Rt3MT1pHE |
| | (1) moves with less frequency | https://youtu.be/lJDfz7KqApY |
| | (2) moves constantly, restless | https://youtu.be/cfET4CN4g0w |
| | (3) reluctant to move or does not move | https://youtu.be/X7_uDln8ih0b |
| **Appetite** | (0) normorexia | https://youtu.be/HWymAEgtaO4 |
| | (1) hyperexia | https://youtu.be/QJ4z-TqDnjw |
| | (2) hyporexia | https://youtu.be/SNgFH5Yt-1A |
| | (3) anorexia | https://youtu.be/pYWA1VwSHYo |
| **Attention to the affected area** | A. elevates pelvic limb or alternates the support of the pelvic limb | https://youtu.be/ndrx0h_nc-Y |
| | B. scratches or rubs the painful area | https://youtu.be/qVkDWKdTjEk |
| | C. moves and/or runs away and/or jumps after injury of the affected area | https://youtu.be/RV0c3bIFfdc |
| | D. sits with difficulty | https://youtu.be/Qq0e1CbRQYU |
| | (0) all the above behaviours are absent | |
| | (1) presence of one of the above behaviours | |
| | (2) presence of two of the above behaviours | |
| | (3) presence of three or all the above behaviours | |
| **Miscellaneous behaviours** | A. wags tail continuously and intensely | https://youtu.be/cfrD0bN5BK4 |
| | B. bites the bars or objects | https://youtu.be/xyw9O14h9dg |
| | C. the head is below the line of the spinal column. | https://youtu.be/qKQRqY0hCY4 |
| | D. presents difficulty in overcoming obstacles (example: other animal) | https://youtu.be/6ucHv8245N4 |
| | (0) all the above behaviours are absent | |
| | (1) presence of one of the above behaviours | |
| | (2) presence of two of the above behaviours | |
| | (3) presence of three or all the above behaviours | |

Complete play list: https://www.youtube.com/watch?v=-loODGUwmS0&list=PLTDt73d-ilJMHnzJdkzlA8h8Fl2iMeTSR

**Table 4. Intra-observer reliability or agreement (confidence interval) for each item of the UPAPS.**

| Items of the scale* | Observers | | | |
|---|---|---|---|---|
| | Gold standard | Evaluator 1 | Evaluator 2 | Evaluator 3 |
| Posture | **0.8** (0,72–0.87) | **0.79** (0.70–0.87) | **0.64** (0.49–0.78) | **0.65** (0.48–0.82) |
| Interaction | **0.69** (0.57–0.80) | **0.78** (0.69–0.86) | **0.66** (0.52–0.81) | **0.68** (0.58–0.79) |
| Activity | **0.62** (0.51–0.74) | **0.63** (0.51–0.75) | **0.68** (0.53–0.83) | **0.67** (0.55–0.80) |
| Appetite | **0.62** (0.42–0.82) | **0.78** (0.64–0.91) | **0.70** (0.49–0.90) | **0.76** (0.63–0.89) |
| Attention to the area | **0.72** (0.63–0.81) | **0.63** (0,52–0,75) | **0.67** (0.55–0.78) | **0.54** (0.40–0.68) |
| Miscellaneous | **0.66** (0.55–0.76) | **0.69** (0.60–0.78) | **0.70** (0.59–0.80) | **0.54** (0.40–0.69) |
| **Total score**** | **0.88** (0.84–0.91) | **0.85** (0.80–0.89) | **0.79** (0.72–0.84) | **0.82** (0.76–0.86) |

UPAPS: UNESP-Botucatu pig composite acute pain scale. The weighted Kappa coefficient was used to calculate the correlation of each item of the scale, encompassing all moments of the 2nd assessment phase. For the total score of UPAPS, the intraclass correlation coefficient type "consistency" was used, and its 95% CI. Interpretation: 0.81–1.0 = very good; 0.61–0.80 = good; 0.41–0.60 = moderate; 0.21–0.40 = reasonable; < 0.20 = poor(38).

affected area and miscellaneous behaviours showed responsiveness except scratching or rubbing the painful area, biting the bars, and difficulty in overcoming obstacles.

## Specificity and sensitivity

Specificity was excellent for all items (> 95%). Sensitivity was excellent for posture and activity and good for attention to the affected area and miscellaneous behaviours. Interaction and interest in the surroundings and appetite did not demonstrate sensitivity (Table 13).

**Table 5. Matrix intraclass coefficient (confidence interval 95%) of the sum of the scores of the UPAPS.**

| | Observer 1 | Observer 2 | Observer 3 |
|---|---|---|---|
| **Gold standard** | **0.81** (0.76–0.86) | **0.80** (0.74–0.85) | **0.62** (0.52–0.70) |
| **Observer 1** | | **0.71** (0.63–0.77) | **0.57** (0.47–0.66) |
| **Observer 2** | | | **0.75** (0.68–0.81) |

UPAPS: UNESP-Botucatu pig composite acute pain scale. Interpretation: 0.81–1.0 = very good; 0.61–0.80 = good; 0.41–0.60 = moderate; 0.21–0.40 = reasonable; < 0.20 = poor(38).

**Table 6. Inter-observer reliability (confidence interval) between the gold standard and other observers of the UPAPS.**

| Items of the scale* | Gold standard *vs* | | |
|---|---|---|---|
| | Observer 1 | Observer 2 | Observer 3 |
| Posture | **0.66** (0.55–0.76) | **0.51** (0.62–0.74 | **0.44** (0.32–0.55) |
| Interaction | **0.75** (0.65–0.85) | **0.71** 0.59–0.83) | **0.53** (0.38–0.67) |
| Activity | **0.53** (0.40–0.65) | **0.69** (0.58–0.81 | **0.53** (0.42–0.64) |
| Appetite | **0.61** (0.43–0.80) | **0.76** (0.60–0.91) | **0.55** (0.34–0.76) |
| Attention to the area | **0.64** (0.53–0.76) | **0.63** (0.52–0.74) | **0.36** (0.24–0.48) |
| Miscellaneous | **0.62** (0.53–0.71) | **0.57** (0.46–0.68) | **0.43** (0.34–0.53) |
| **Total score**** | **0.81** (0.76–0.86) | **0.80** (0.74–0.85) | **0.62** (0.52–0.70) |

UPAPS: UNESP-Botucatu pig composite acute pain scale.
*The weighted Kappa coefficient was used to calculate the correlation of each item of the scale. For the total score of UPAPS, the intraclass correlation coefficient type "consistency" was used, and its 95% CI. The agreement between the gold standard evaluator and other observers was assessed encompassing all moments of the 2nd assessment phase. Interpretation: 0.81–1.0 = very good; 0.61–0.80 = good; 0.41–0.60 = moderate; 0.21–0.40 = reasonable; < 0.20 = poor(38).

**Table 7. Spearman correlation between the UPAPS and the Visual Analogue Scale (VAS), Numerical Rating Scale (NRS), and Simple Descriptive Scale (SDS).**

| Scale | Observers | | | | |
|---|---|---|---|---|---|
| | Gold-standard | Evaluator 1 | Evaluator 2 | Evaluator 3 | All |
| VAS | 0.91 | 0.89 | 0.78 | 0.83 | 0.85 |
| NRS | 0.93 | 0.90 | 0.83 | 0.83 | 0.88 |
| SDS | 0.91 | 0.88 | 0.78 | 0.83 | 0.85 |

UPAPS: UNESP-Botucatu pig composite acute pain scale. P value = 0.

**Table 8. Percentage of pigs for which rescue analgesia was indicated according to clinical experience and according to the Youden index of the UPAPS.**

| Observers | Gold | | 1 | | 2 | | 3 | | All | |
|---|---|---|---|---|---|---|---|---|---|---|
| RA | Exp | YI | Exp | YI | Exp | YI | Exp | YI | Exp | YI |
| M1 | 0 | 0 | 0 | 11 | 0 | 2 | 0 | 9 | 0 | 6 |
| M2 | 96 | 98 | 84 | 91 | 47 | 76 | 47 | 71 | 68* | 84* |

Calculation based on 45 pigs for each observer and 180 pigs for all evaluators. RA—indication of rescue analgesia according to clinical experience scored at the end of each video analysis (Exp) and according to the Youden index of the UPAPS (score ≥ 4). UPAPS—UNESP-Botucatu pig composite acute pain scale. Gold—gold standard observer; M1—preoperative; M2—postoperative, before rescue analgesia. Youden index ≥ 4 is representative of the cut-off point for indication of rescue analgesia (see Tables 15 and 16 and Fig 5 for results of Youden index).
* expresses difference between Exp and YI according to Fisher's Exact Test (p = 0.0008).

## Distribution of scores

The range of scores was distributed as expected according to pain degree (Figs 3 and 4). Scores 0 predominated before surgery (M1) as expected because pigs were not suffering pain. Scores 1, 2, and 3 predominated at 4 and 24 hours after surgery (M2 and M4), corresponding to the moments of intense and moderate pain. Except for miscellaneous behaviour, only scores of 1

**Table 9. Principal component analysis showing load, eigenvalue, and variance after the analysis of the main components with the Kaiser criterion of the UPAPS.**

| Items of the Scale | Factor Load* Dimension 1 | Factor Load* Dimension 2 |
|---|---|---|
| Posture | **-0.88** | 0.07 |
| Interaction and interest in the surroundings | **-0.82** | -0.20 |
| Activity | **-0.84** | -0.11 |
| Appetite | **-0.50** | **-0.76** |
| Attention to the surgical wound | **-0.75** | 0.38 |
| Miscellaneous | **-0.71** | 0.40 |
| Eigenvalue | **3.47** | 0.94 |
| Variance | **57.88** | 15.59 |

UPAPS: UNESP-Botucatu pig composite acute pain scale. The principal component analysis defines the number of dimensions determined by different variables that establish the scale extension. Two factors, dimensions or domains, were defined for UPAPS.
*Factor load represents the correlation between the items and the factors. The factor loads in bold indicate the variables which contribute to each factor. The factorial structure was confirmed when items showed a factor load ≥ 0.50 or ≤ -0.50 (in bold) (43). The item appetite showed the lowest factor load in dimension 1, but a high factor load in dimension 2, and considering that, in the ethogram, time spent eating was lower at the moment of pain compared to the moment after rescue analgesia, this item was maintained in the scale. Please see Fig 2 for a graphical example.

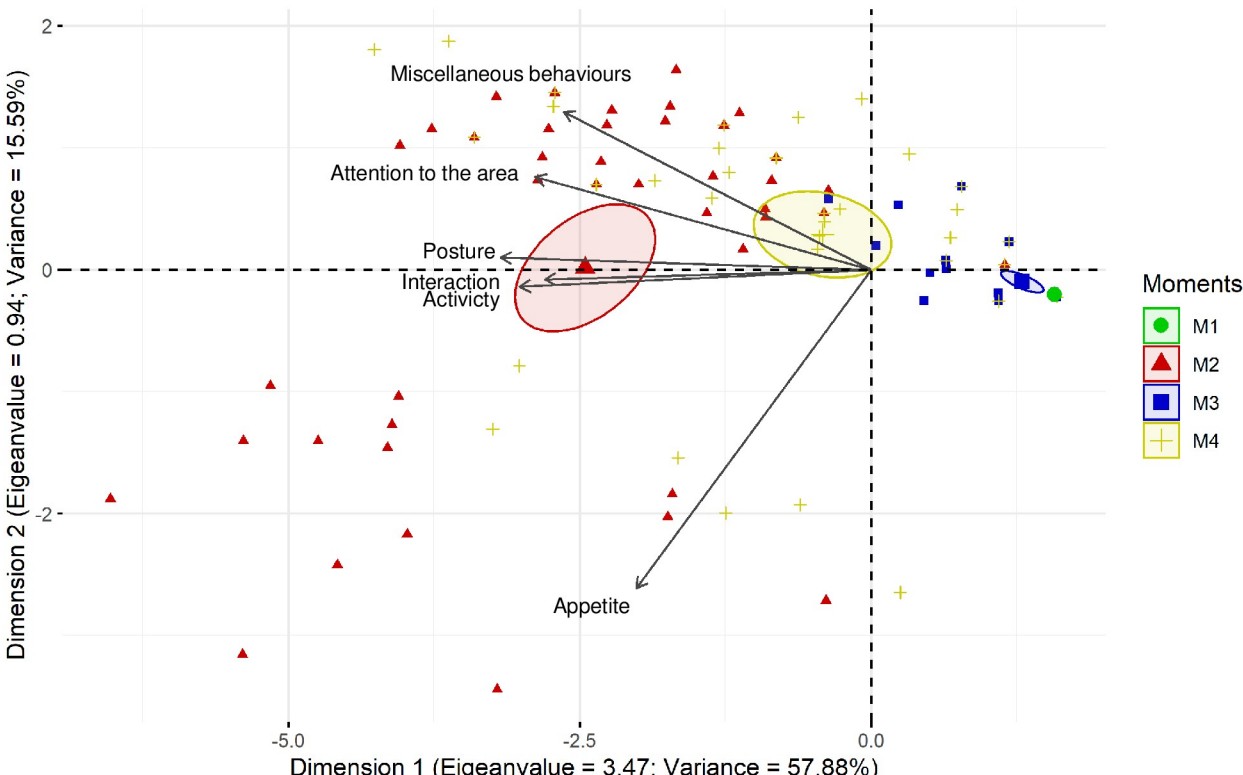

**Fig 2. Principal component analysis of the items of the UPAPS.** UPAPS—UNESP-Botucatu pig composite acute pain scale. Data were obtained from the gold standard observer analysis at four evaluation moments. Each number corresponds to one pig and each colour corresponds to one moment (M1—green, pigs 1 to 45; M2—red, pigs 46 to 90, correspond to pigs from 1 to 45 of M1; M3—blue, pigs 91 to 135, correspond to pigs from 1 to 45 of M1 and M4—yellow, pigs 136 to 180, correspond to pigs from 1 to 45 of M1). The ellipses were constructed according to the pain assessment moments (M2—red, M3—blue, and M4—yellow). The ellipse corresponding to the moments pigs were suffering intense (M2) and moderate pain (M4) are positioned at the left side of the figure. On the opposite side are the ellipses corresponding to the moments pigs were probably not suffering pain (M1 and M3). All items on the scale were influenced by pain (M2 and M4) since their vectors are positioned in the direction of these ellipses.

were present after rescue analgesia (M3), corresponding to mild pain and reflecting pain reduction after analgesia. The number of scores of 3 was greater at M2 than at M4, confirming that pain was the most intense after surgery and before rescue analgesia (M2).

## ROC Curve for determination of the cut-off point of the UPAPS

The cut-off score for pain diagnosis and recommendation for analgesia determined by the Youden index at the 2nd phase of video analysis was $\geq 4$, $\geq 6$, $\geq 4$, and $\geq 5$ for the gold standard, 1, 2, and 3 evaluators respectively, and $\geq 4$ for all grouped observers within the scoring margin of 0–18 points (Fig 5 and Table 14). When considering all observers grouped, the interval between the sensitivity and specificity values of 0.90 was between 3.7 and 4.7 (individual values ranged between 2.4 and 5.6). Resampling confidence interval (Bootstrap) for the Youden index was between 3.5 and 4.5 (individual values ranged between 2.5 to 5.5). The calculated diagnostic uncertainty zone of all observers suggests that the threshold score for detecting truly positive (really suffering pain) pigs is $\geq 6$, whereas the cut-off score for detecting truly negative (painless) pigs is $\leq 3$. Area under the curve (AUC) values of all evaluators

**Table 10. Item-total correlation between each item score and total score for the UPAPS.**

| Item | MA |
|---|---|
| *Excluding each item below* | |
| Posture | **0.87** |
| Interaction and interest in the surroundings | **0.64** |
| Activity | **0.83** |
| Appetite | **0.42** |
| Attention to the surgical wound | **0.79** |
| Miscellaneous | **0.73** |

UPAPS: UNESP-Botucatu pig composite acute pain scale. Interpretation of Spearman's item-total correlation coefficient: Values above 0.3 were considered acceptable (23). MA—data from moments grouped (M1 + M2 + M3 + M4).

**Table 11. Internal consistency of the UPAPS.**

| Item | MA |
|---|---|
| Full scale | **0.89** |
| *Excluding each item below* | |
| Posture | **0.84** |
| Interaction and interest in the surroundings | **0.87** |
| Activity | **0.86** |
| Appetite | **0.90** |
| Attention to the surgical wound | **0.86** |
| Miscellaneous | **0.88** |

UPAPS: UNESP-Botucatu pig composite acute pain scale. Cronbach's α coefficient was calculated for the total score and by excluding each item of the scale. Values for the α coefficient were regarded as follows: 0.60–0.64, minimally acceptable; 0.65–0.69, acceptable; 0.70–0.74, good; 0.75–0.80, very good; and > 0.80, excellent (45). MA—data from moments grouped (M1 + M2 + M3 + M4).

grouped was 0.98 (individual values ranged between 0.96–1), pointing out that the tool has excellent discriminatory ability (Fig 5).

The cut-off points for rescue analgesia of unidimensional scales defined by the ROC curve and the Youden index were $\geq 3$ for NRS, $\geq 2$ for SDS and $\geq 33$ for VAS (Table 15).

**Table 12. Medians (range) of the total scores (0–18) of the UPAPS assessed at the 2nd phase of video analysis.**

| Moments | Observers | | | |
|---|---|---|---|---|
| | Gold-standard | Evaluator 1 | Evaluator 2 | Evaluator 3 |
| M1 | 0 (0–1)[c] | 0 (0–6)[c] | 0 (0–6)[b] | 0 (0–5)[c] |
| M2 | 8 (1–17)[a] | 6 (0–16)[a] | 5 (0–14)[a] | 6 (0–16)[a] |
| M3 | 0 (0–4)[c] | 1 (0–12)[c] | 0 (0–4)[b] | 0 (0–3)[c] |
| M4 | 4 (0–12)[b] | 3 (0–13)[b] | 1 (0–9)[b] | 2 (0–102)[b] |

UPAPS: UNESP-Botucatu pig composite acute pain scale. Differences in pain scores for each observer between moments, being a > b > c according to the Friedman test (p < 0.05).

**Table 13. Specificity (in M1) and sensitivity (in M2) of each item of the UPAPS assessed according to the 2$^{nd}$ phase of the gold standard video analysis.**

| Items | Specificity (%)[1] | Sensitivity (%)[2] |
|---|---|---|
| Posture | 100 | 98 |
| Interaction and interest in the surroundings | 100 | 62 |
| Activity | 100 | 96 |
| Appetite | 100 | 31 |
| Attention to the affected area | 100 | 89 |
| Miscellaneous | 96 | 93 |

UPAPS: UNESP-Botucatu pig composite acute pain scale.

[1]Gold standard observer scores recorded at M1 (preoperative) were changed to dichotomous items. Specificity = TN/TN+FP, where TN = number of true negatives (score 0); FP = number of false positives (scores 1, 2, or 3).

[2]Gold standard observer scores recorded at M2 (postoperative, before rescue analgesia) were changed to dichotomous items. Sensitivity = TP/TP+FN, where TP = number of true positives (scores 1, 2 or 3), FN = number of false negatives (score 0). Specificity and sensitivity were classified as excellent (95–100%), good (85–94,9%), moderate (70–84,9%), or non-sensitive or non-specific (<70) [29].

The mean percentage of pigs showing scores inside the diagnostic uncertainty zone was 10%. According to each observer at M2, from 13 to 29% of pigs would fit in the diagnostic uncertainty zone (Table 16).

Total UPAPS scores were separated into four grades of pain intensity (Fig 6). Median (range) values were: no pain (1; 0–2), mild (5; 3–6), moderate (8; 7–9) and intense pain (12; 10–17) (Fig 7). Based on this grading, in the moment of most intense pain (M2), considering the results of all observers grouped 11% of pigs were classified as no pain and 37% as mild, 28% as moderate and 24% as intense pain.

## Discussion

Pigs are submitted to painful procedures in rural and research environments, usually without proper anaesthesia and/or analgesia, possibly due to the absence of an easy-to-use and valid instrument to assess pain in this species [17]. This gap has been filled here, as this study validated a scale to assess postoperative pain in pigs according to required validation criteria [52,53].

The ethogram was relevant to identify and/or confirm pain-related behaviours in pigs undergoing castration and supported the construction of the pain scale. The video recordings provided data to validate the instrument to be applied both in farms and research facilities. The proposed pain scale showed good or very good intra and inter-observer agreement, content, construct and criterion validity, adequate item-total correlation, excellent internal consistency, responsiveness, and specificity, good sensitivity and a defined cut-off point for analgesic intervention [20–23].

The adaptation of the piglets to the in-person researcher was apparently opportune. Pigs have the ability to recognize individuals from the first weeks of life and, when socialized with humans, their fear reduces [54,55]. The period of adaptation to the in-person researcher probably contributed to make the pain-related behavioural changes of the pigs more apparent. Appetite increased after rescue analgesia. Overall, normal behaviours decreased, and abnormal behaviours increased after surgery, before rescue analgesia. Extensive literature is available regarding pain-related behaviours in pigs [6,8,32,33,56] and it is beyond the scope of this study

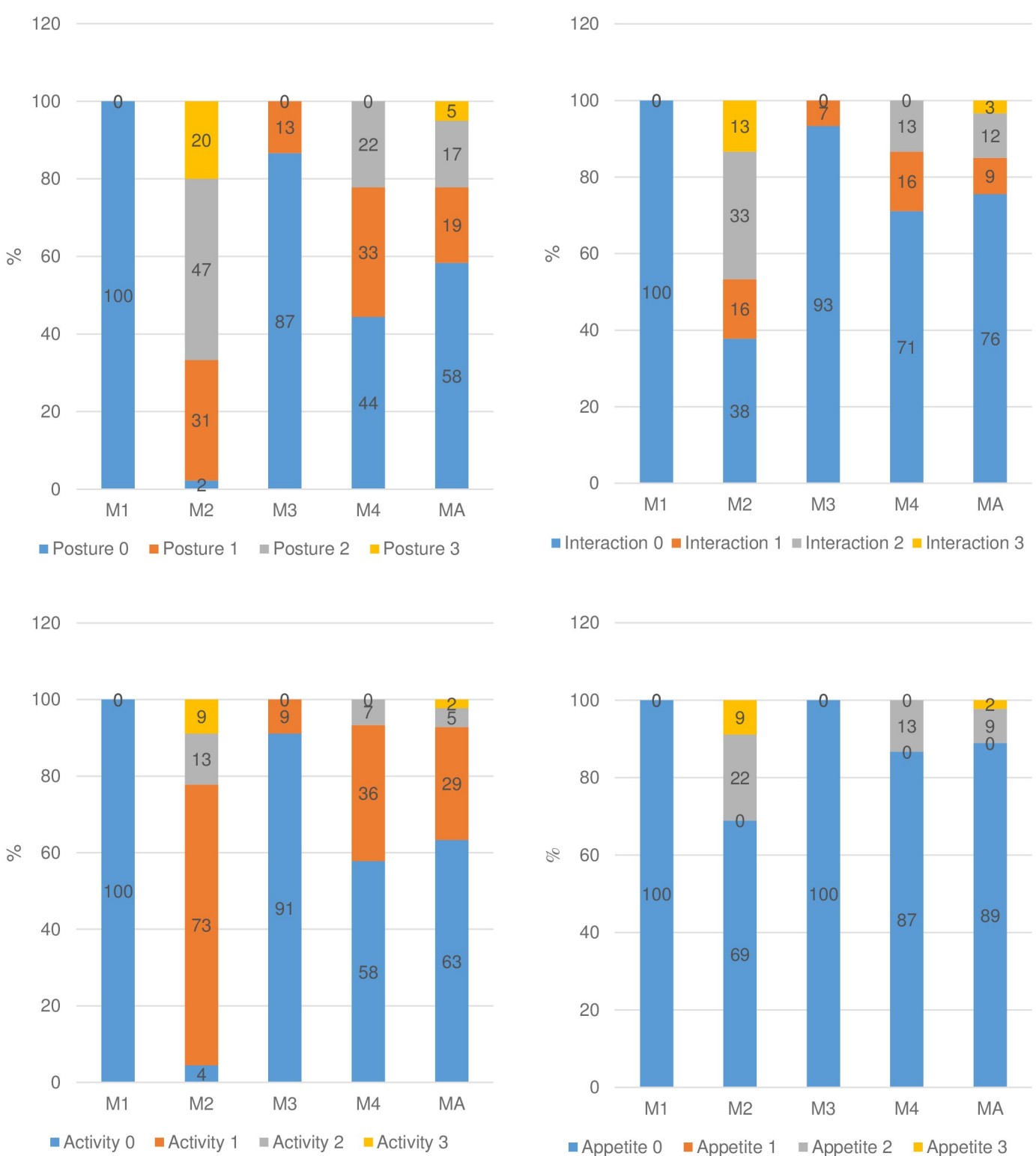

**Fig 3. Frequency of the presence of scores of each item of the UPAPS.** Legend: posture (top left), interaction and interest in the surroundings (top right), activity (bottom left) and appetite (bottom right).

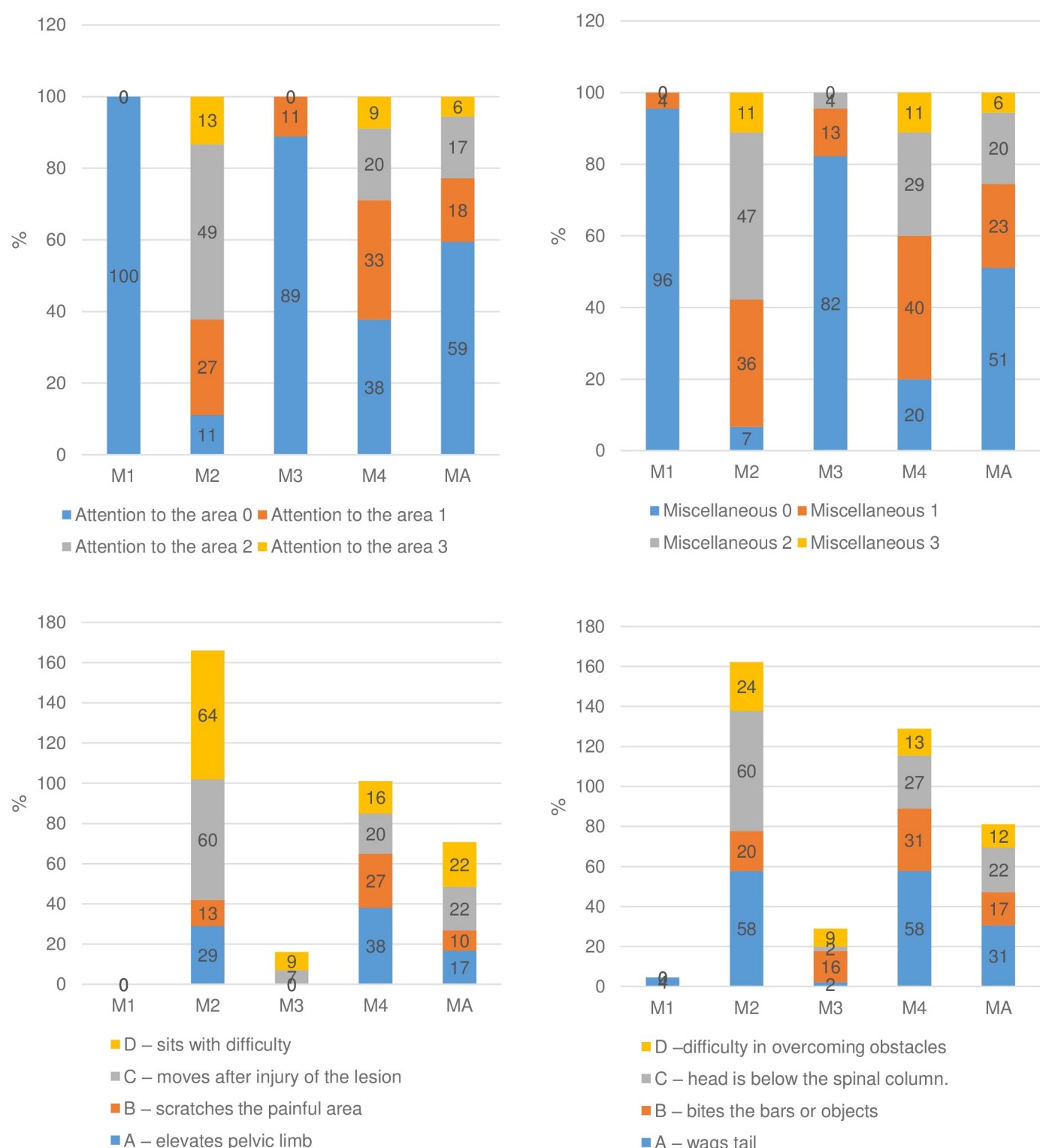

**Fig 4. Frequency of the presence of scores of each item of the UPAPS.** Legend: sum of the scores (top left) and individual scores (bottom left) of attention to the affected area and sum of the scores (top right) and individual scores (bottom right) of miscellaneous behaviours.

(a)

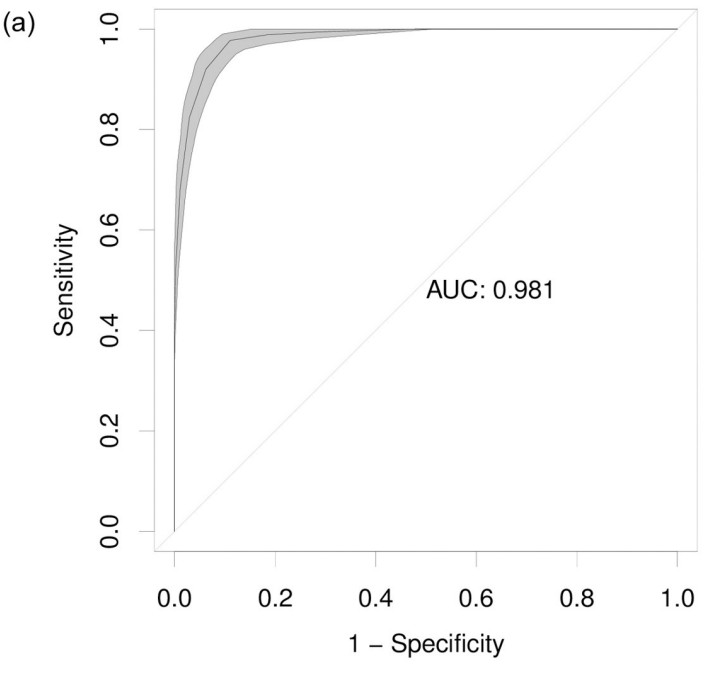

(b)

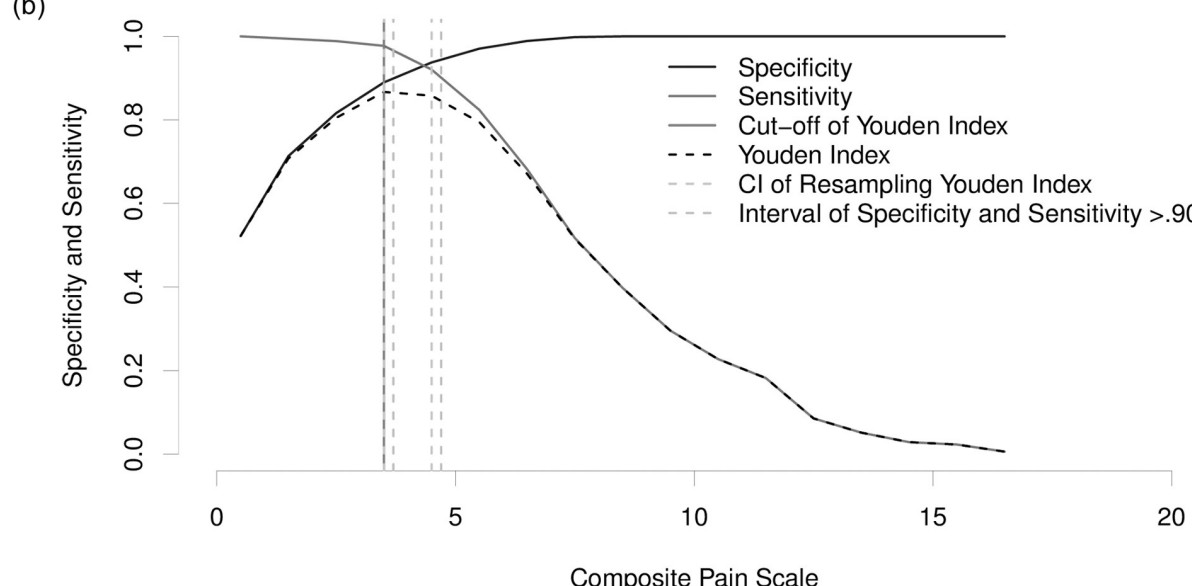

**Fig 5. ROC curve and AUC (left) and two-graphic ROC curve with the diagnostic uncertainty zone for the UPAPS (right).** Legend: UPAPS—UNESP-Botucatu pig composite acute pain scale. Receiver Operating Characteristic (ROC) curve with 95% confidence interval (CI) calculated from 1000 replications and area under the curve (AUC) (A—left). Two-graphic ROC curve, CI of the 1000 replications and of sensitivity and specificity > 0.90 applied to estimate the diagnostic uncertainty zone of the cut-off point using all moments of pain assessment scored by all observers in the 2nd evaluation phase, according to the Youden index for the UPAPS (B—right).

to perform an in-depth analysis of these behaviours. The ethogram was performed to confirm, under our experimental conditions, the behaviours described in other facilities and provide additional supporting data to build the pain assessment instrument. It was clear that surgical castration promoted pain-related behavioural changes, even considering that piglets were

**Table 14. Specificity, sensitivity, and Youden index of each observer for each scale score of the UPAPS (A); 95% confidence intervals of 1000 replications and of sensitivity and specificity > 0.90 applied to estimate the diagnostic uncertainty zone of the cut-off point of each evaluator, according to the Youden index (B).**

| A | Gold standard observer | | | Observer 1 | | | Observer 2 | | | Observer 3 | | | All observers | | |
|---|---|---|---|---|---|---|---|---|---|---|---|---|---|---|---|
| Score | Spec | Sens | Youden index | Spec | Sens | Youden index | Spec | Sens | Youden index | Spec | Sens | Youden index | Spec | Sens | Youden index |
| **4** | **0.96** | **1** | **0.96** | 0.89 | 1 | 0.89 | **0.86** | **0.94** | **0.80** | 0.87 | 0.91 | 0.78 | **0.89** | **0.98** | **0.87** |
| **5** | 0.99 | 0.91 | 0.90 | 0.93 | 0.96 | 0.89 | 0.91 | 0.88 | 0.79 | **0.93** | **0.91** | **0.84** | 0.94 | 0.92 | 0.86 |
| **6** | 1 | 0.77 | 0.77 | **0.98** | **0.91** | **0.90** | 0.94 | 0.81 | 0.75 | 0.97 | 0.78 | 0.75 | 0.97 | 0.82 | 0.79 |
| B | Confidence interval | | | Confidence interval | | | Confidence interval | | | Confidence interval | | | Confidence interval | | |
| | Est | Min | Max | Est | Min | Max | Est | Min | Max | Est | Min | Max | Est | Min | Max |
| **Replication CI** | -- | 3.5 | 4.5 | -- | 3.5 | 5.5 | -- | 3.5 | 5.5 | -- | 2.5 | 5.5 | 3.5 | 3.5 | 4.5 |
| **CI of sens and spec > 0.90** | -- | 2.4 | 4.55 | -- | 3.8 | 5.6 | -- | 4.1 | 4.3 | -- | 4 | 4.6 | -- | 3.7 | 4.7 |
| **AUC** | 01 | 0.99 | 1 | 0.99 | 0.99 | 1 | 0.96 | 0.93 | 0.99 | 0.97 | 0.95 | 1 | 0.98 | 0.97 | 0.99 |
| **Cut-off point** | 3.5 | -- | -- | 5.5 | -- | -- | 3.5 | -- | -- | 4.5 | -- | -- | 3.5 | 3.5 | 4.5 |

UPAPS—UNESP-Botucatu pig composite acute pain scale; Spec—specificity; Sens—sensitivity; CI—confidence interval; AUC—area under the curve; Est—estimated; Min—minimum; Max—maximum

**Table 15. Scores, specificity, sensitivity, and Youden index corresponding to rescue analgesia indication of the UPAPS and unidimensional scales.**

| Scales | Score≥ | Diagnostic uncertainty zone | Specificity | Sensitivity | Youden index |
|---|---|---|---|---|---|
| **UPAPS** | 4 | 3–6 | 0.89 | 0.98 | 0.87 |
| **NRS (1–10)** | 3 | 2–4 | 0.94 | 0.98 | 0.91 |
| **SDS (0–3)** | 2 | 1–2 | 0.97 | 0.92 | 0.89 |
| **VAS** | 33 | 21–39 | 0.96 | 0.97 | 0.92 |

UPAPS—UNESP-Botucatu pig composite acute pain scale; NRS—numerical rate scale; SDS—simple descriptive scale; VAS—visual analogue.

castrated with local anaesthesia. This finding reinforces the need for proper use of anaesthetics and analgesics in the peri-operative period in pigs [56].

This study showed that the UPAPS was reliable and valid to evaluate postoperative pain after castration in pigs. Reliability indicates the precision of the tool based on the reproducibility [57]. Evaluators' experience improves reliability [58] and instruction and training improved the recognition of pain in laboratory animals [59]. The observers of this study had experience

**Table 16. Percentage of pigs present in the diagnostic uncertainty zone according to the Youden index of the UPAPS (scores 4 or 5).**

| Observer (number of pigs) | Gold (45) | 1 (45) | 2 (45) | 3 (45) | All (180) |
|---|---|---|---|---|---|
| **M1** | 0 | 7 | 2 | 2 | 3 |
| **M2** | 22 | 13 | 18 | 29 | 21 |
| **M3** | 2 | 0 | 0 | 2 | 1 |
| **M4** | 20 | 18 | 13 | 09 | 15 |
| **MA** | 11 | 9 | 8 | 11 | 10 |

UPAPS—UNESP-Botucatu pig composite acute pain scale; Gold—gold standard observer; M1—preoperative; M2—postoperative, before rescue analgesia; M3—postoperative, after rescue analgesia; M4—24h postoperative; MA—data of grouped moments (M1 + M2 + M3 + M4).

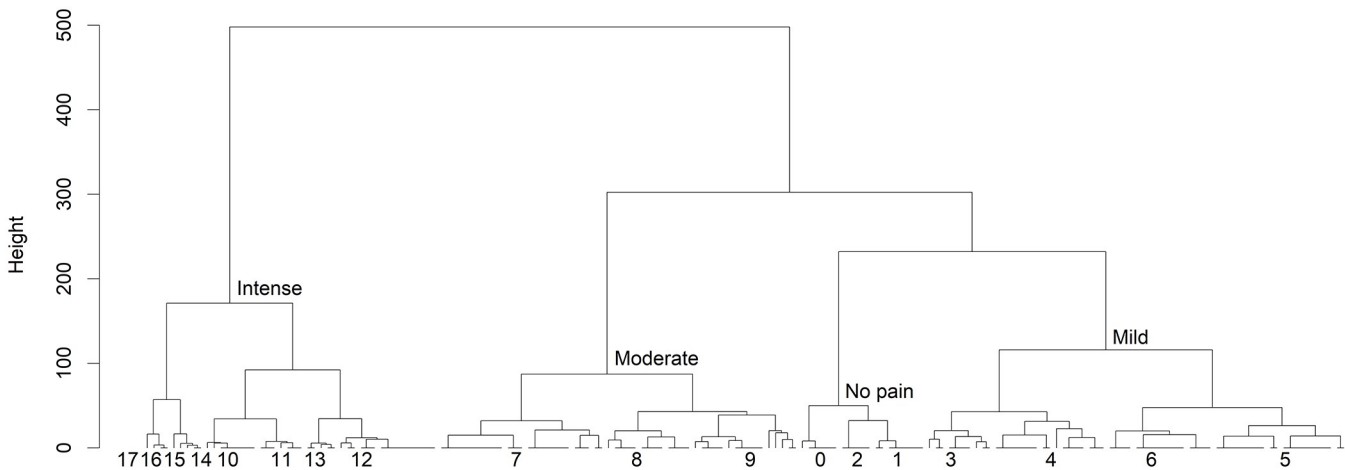

**Fig 6. Dendogram created by the non-hierarchical cluster analysis based on the total score of the UPAPS.** Legend: UPAPS—UNESP-Botucatu pig composite acute pain scale. The scores were graded into 4 groups: no pain (0–2), mild (3–6), moderate (7–9) and intense pain (10–17).

in pain assessment in other species, but not in pigs, and they were untrained before the observations. The inter and intra-observer agreement results guarantees the scale´s efficiency to provide consistent results among observers and when the instrument is assessed twice by the same observer respectively [23].

Validity indicates that the instrument can precisely quantify what is proposed. Validity composes the 'three Cs': content, criterion, and construct validities [23].

The content validity guarantees that the tool approaches the essential domains or embraces all aspects of the variable to be measured [23]. This was grounded by the appreciation of experienced veterinarians in the field, who analysed the representativeness of each item in relation to the scale as a whole [22,37], the ethogram findings and pain-related behaviours described in the literature [6,8,32,33,56].

The systematic evaluation of the tool should be carried out through comparison with a pre-existing external criterion which has already been validated and is therefore considered the gold-standard [60]. Considering the non-existence of a valid tool to assess pain in pigs, the same methodology applied to cats [21] and cattle [22] was adopted and concurrent criterion validity was based on the agreement between the pain scores attributed by the evaluators and a gold-standard evaluator, intensely involved in the planning and execution of the experimental part of the study. The excellent correlation between the scores of the proposed scale *versus* the VAS, NRS, and SDS, confirmed the concurrent criterion validity of the scale. Although these scales have not been validated in pigs, they have been used to assess pain in veterinary medicine [21,22,61].

Predictive criterion validity was established by the finding that according to the Youden index, 84% of the pigs would be treated with rescue analgesia at the moment of the most intense pain [M2]; therefore, the instrument predicted well that pigs were experiencing pain at this moment and improved decision making when compared to clinical experience, especially for observers 2 and 3.

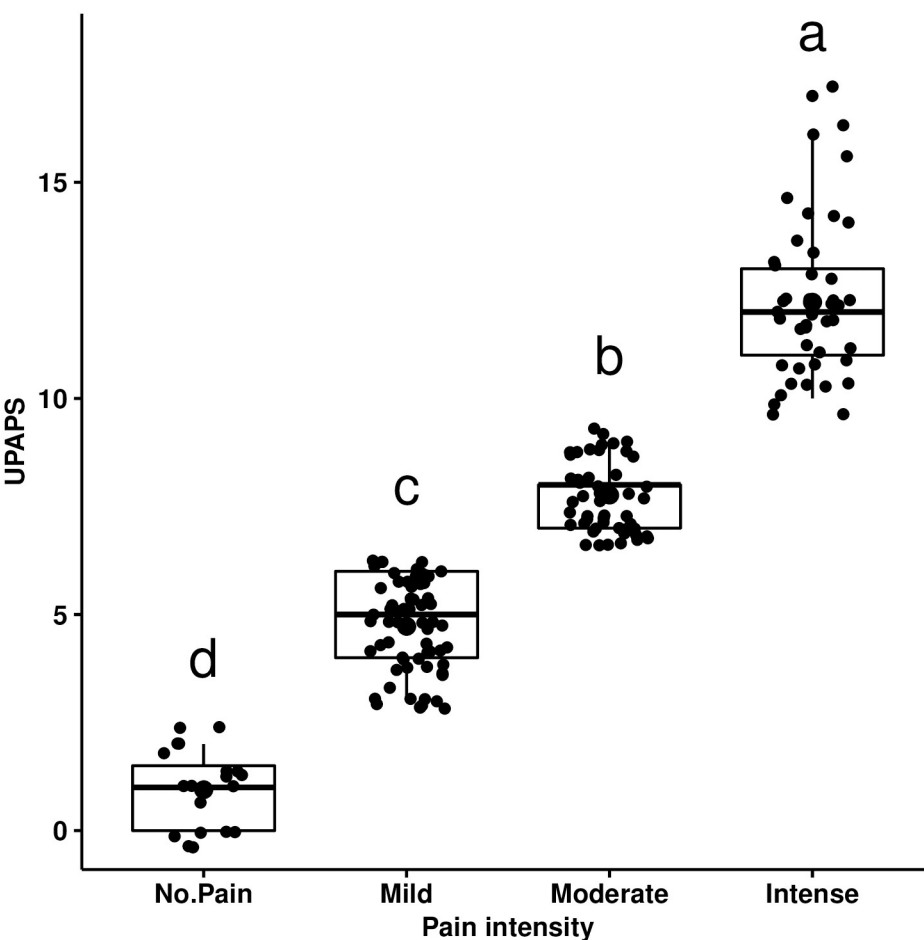

**Fig 7. Box plot of the scores related to pain intensity of the UPAPS.** Legend: UPAPS—UNESP-Botucatu pig composite acute pain scale. Data were grouped at the time of most intense pain (M2). According to the median, minimum and maximum scores obtained by the non-hierarchical cluster analysis, the scores were graded into 4 groups: no pain (0–2), mild (3–6), moderate (7–9) and intense pain (10–17). Different letters indicate statistical difference according to the Kruskal-Wallis test, with a> b> c > d.

The proposed scale generated only one factor and, therefore, from a statistical point of view, it is considered unidimensional, like in cattle [22]. The dimensions and domains of the tool were determined by principal component analysis, which is normally used to relate the tool's variables in a grouped manner [62]. An instrument is multidimensional when it includes other pain experiences, rather than only intensity. These features comprise quality and temporal characteristics, such as sensory, affective, and cognitive dimensions. Regarding pain, these three dimensions show the same trend and therefore, a high correlation. A classic example of a multidimensional scale is the McGill Pain Questionnaire that includes sensory, affective, evaluative, and miscellaneous components [63,64], however the number of its dimensions vary between different studies, patient categories, and statistical models used [65]. It is precocious to make assumptions about the dimension of the instrument presented here, as only one type of pain was assessed (soft tissue) and only one statistical model performed, however different biological "dimensions" of pain are included in the pig scale, such as physiologic (appetite), sensory (posture, activity, attention to the affected area, and miscellaneous), affective

(interaction and interest in the surroundings), and temporal (response to analgesia). Future studies using different pain models and statistical approaches may either confirm or not if the scale is multidimensional.

The primary criteria of principal component analysis was to reject items with a correlation coefficient < 0.5 [43]. Appetite was maintained as it was present in dimension 2 and, according to the ethogram, the time spent eating was lower when pigs were suffering pain. Hyporexia is classically considered an important behaviour in animals experiencing pain [10].

All items showed acceptable item–total correlation, indicating their individual relevance, and assuring scale homogeneousness [23]. The internal consistency was excellent, like in cattle [22]. This variable is another measure of reliability of the tool [57] and ensures that the scores of the items that compose the scale may be summed, to guarantee that the total score is representative of pain intensity [21]. The values of items were equivalent, indicating that they have similar tendencies [41].

Construct validity is tested by the responsiveness of the scale. The instrument was able to differentiate degrees of pain, such as mild/moderate and intense, and therefore detected the predictable alterations in the construct [57]. The expected result that the use of analgesics would reduce pain was confirmed by the reduction in scores after rescue analgesia [21,22]. It was remarkable that pain was present again at 24 hours postoperatively. By this time the effect of the analgesics had abated, as the elimination half-life of flunixin and morphine is 7.5 hours [5.6–13 hours] [66] and 65 minutes respectively [67].

Like in cats [21] and cattle [22], the analysis of the ROC curve determined the score required for the analgesic intervention in pigs [47]. The high AUC indicated that the scale presents optimum accuracy, in other words, the tool correctly qualifies individuals with and without pain [46,47]. Similar results were found in the validation of pain scales in cats [21] and cattle [22], using the same methodology.

The definition of the scores which suggest the need for analgesics to control pain, substantiates and increases the confidence in decision-making for pig specialist veterinarians and researchers who are not confident about pain assessment and treatment [2, 3]. Pain mitigation guarantees both the pig's welfare and improves productive performance [10] in the rural environment and may avoid short and long term hyperalgesia, which could compromise scientific results [68], based on the principle that "happy animals make good science" [69].

The cut-off score for determining pain for analgesic recommendation varied among raters from $\geq 4$ to $\geq 6$. This could be attributed to their experience, nevertheless, all raters presented high accuracy of the test. Considering the diagnostic uncertainty zone of all raters grouped, it can be assumed that animal rating scores $\geq 6$ of a total of 18 points, correspond to pigs that are truly in pain (true positives) and those with scores $\leq 3$ are true negatives and without pain. The adoption of this methodology improves the accuracy of pain diagnosis and provision of analgesia. However, even when the scores are < 6, the discernment between using or not using analgesics must always be grounded on observers' decisions. This was the case of 10% of pigs in the moment of the most intense pain.

So far, to our knowledge there is no information in the literature about the calculation of pain-intensity ranking in animals, except empirically [24]. This data is relevant for the classification of pain intensity in different types of surgery and for decision making for selecting analgesic protocols. The upper limit of mild pain scores and the diagnostic uncertainty zone was the same (6), warranting that pigs undergoing moderate pain should receive analgesia.

The main limitations of this study were that the instrument was validated only for a specific type of surgery (orchiectomy), in piglets, and by using video analysis. Although castration is probably the most common procedure producing intense pain in swine in the rural environment, further studies are demanded to test this tool in different painful procedures, like

orthopaedic surgery, as well as in clinical circumstances, to ensure its versatility. Another limitation is that it is not possible to extrapolate our results to pigs of different ages. Although in pre-weaning pigs ranging from 3 to 17 days of age undergoing castration, there were no age-dependent differences in pain-related behaviours [8,70], pain-related behaviour changes were more evident in older pigs aged 7 weeks than those aged 2 weeks [71]. In the present study, the pigs were 38 days old, which is an intermediate age between pre-weaning and adult life and this was a similar age to pigs that display clearer pain-related behaviours compared to younger pigs [71]. A third limitation is that video analysis is not the same as in-person analysis. Observation by video may miss some details observed in real-time, otherwise avoids inhibition of disturbance behaviour caused by the presence of the observer as reported in horses [72]. Video analysis provides material to develop the ethogram, may be reviewed and watched as many times as necessary and is an important step to validate a scale, as reported in cats [37], cattle [22], and horses [27], especially to assess intra-observer reliability. According to our previous study in cats, results from video analysis were also reproducible in a clinical scenario [21,37].

## Conclusion

The UNESP-Botucatu composite scale for assessing post-operative pain in pigs (UPAPS) is a valid, reliable, responsive, specific, and sensitive tool, with excellent internal consistency and discriminatory ability, and may be used as an instrument to assess acute pain in pigs submitted to orchiectomy. The well-defined cut-off point supports the evaluator´s decision to provide or not analgesia.

## Supporting information

**S1 Table. Medians (range) of the score of each item of the UPAPS assessed at the 2nd phase of video analysis by the gold standard observer.**
(DOCX)

**S1 Data. Data of the pigs.**
(XLSX)

**S1 Fig.**
(JPG)

## Author Contributions

**Conceptualization:** Stelio Pacca Loureiro Luna, Ana Lucélia de Araújo.

**Data curation:** Stelio Pacca Loureiro Luna, Ana Lucélia de Araújo, Felipe Garcia Telles, Pedro Henrique Esteves Trindade.

**Formal analysis:** Stelio Pacca Loureiro Luna, Ana Lucélia de Araújo, Flávia Augusta de Oliveira, Liliane Marinho dos Santos Azerêdo, Pedro Henrique Esteves Trindade.

**Funding acquisition:** Stelio Pacca Loureiro Luna.

**Investigation:** Stelio Pacca Loureiro Luna, Ana Lucélia de Araújo, Pedro Isidro da Nóbrega Neto, Liliane Marinho dos Santos Azerêdo, Felipe Garcia Telles.

**Methodology:** Stelio Pacca Loureiro Luna, Ana Lucélia de Araújo, Juliana Tabarelli Brondani, Flávia Augusta de Oliveira, Liliane Marinho dos Santos Azerêdo, Felipe Garcia Telles, Pedro Henrique Esteves Trindade.

**Project administration:** Stelio Pacca Loureiro Luna, Ana Lucélia de Araújo.

**Resources:** Stelio Pacca Loureiro Luna, Felipe Garcia Telles.

**Software:** Juliana Tabarelli Brondani, Pedro Henrique Esteves Trindade.

**Supervision:** Stelio Pacca Loureiro Luna, Pedro Isidro da Nóbrega Neto.

**Validation:** Stelio Pacca Loureiro Luna, Ana Lucélia de Araújo, Juliana Tabarelli Brondani, Flávia Augusta de Oliveira, Pedro Henrique Esteves Trindade.

**Visualization:** Stelio Pacca Loureiro Luna, Ana Lucélia de Araújo, Flávia Augusta de Oliveira, Liliane Marinho dos Santos Azerêdo, Pedro Henrique Esteves Trindade.

**Writing – original draft:** Stelio Pacca Loureiro Luna, Ana Lucélia de Araújo, Juliana Tabarelli Brondani.

**Writing – review & editing:** Stelio Pacca Loureiro Luna, Ana Lucélia de Araújo, Pedro Isidro da Nóbrega Neto.

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
