## [Decision Letter · Decision Letter 0]

22 Jan 2020

PONE-D-19-30670

Validation of the UNESP-Botucatu composite pain scale for assessing postoperative pain in pigs

PLOS ONE

Dear Prof Stelio Pacca Loureiro Luna,

Thank you for submitting your manuscript to PLOS ONE. After careful consideration, we feel that it has merit but does not fully meet PLOS ONE’s publication criteria as it currently stands. Therefore, we invite you to submit a revised version of the manuscript that addresses the points raised during the review process.

Dear Authors,

it seems that both the reviewer have some minor related to your study. Please provide a complete response or amend the manuscript accordingly their suggestions if possible.

We would appreciate receiving your revised manuscript by Mar 07 2020 11:59PM. To enhance the reproducibility of your results, we recommend that if applicable you deposit your laboratory protocols in protocols.io, where a protocol can be assigned its own identifier (DOI) such that it can be cited independently in the future. For instructions see: http://journals.plos.org/plosone/s/submission-guidelines#loc-laboratory-protocols

We look forward to receiving your revised manuscript.

Kind regards,

Francesco Staffieri

Academic Editor

PLOS ONE

Additional Editor Comments (if provided):

Dear Authors,

it seems that both the reviewer have some minor related to your study. Please provide a complete response or amend the manuscript accordingly their suggestions if possible

Journal Requirements:

2. In your Methods section, please provide additional details regarding the animals used in your study and ensure you have described the source. For more information regarding PLOS' policy on materials sharing and reporting, see https://journals.plos.org/plosone/s/materials-and-software-sharing#loc-sharing-materials.

3. In your Methods section, please include a comment about the state of the animals following this research. Were they euthanized or housed for use in further research? If any animals were sacrificed by the authors, please include the method of euthanasia and describe any efforts that were undertaken to reduce animal suffering.

4. Please amend the manuscript submission data (via Edit Submission) to include author Flávia Augusta de Oliveira.

5. Please ensure that you refer to Figure 2 and 5 in your text as, if accepted, production will need this reference to link the reader to the figure.

6. Please upload a new copy of Figure 5 as the detail is not clear. Please follow the link for more information: http://blogs.PLOS.org/everyone/2011/05/10/how-to-check-your-manuscript-image-quality-in-editorial-manager/

7. We note you have included a table to which you do not refer in the text of your manuscript. Please ensure that you refer to Table 2 in your text; if accepted, production will need this reference to link the reader to the Table.

Reviewers' comments:

Reviewer's Responses to Questions

**Comments to the Author**

1. Is the manuscript technically sound, and do the data support the conclusions?

Reviewer #1: Yes

Reviewer #2: Yes

2. Has the statistical analysis been performed appropriately and rigorously? 

Reviewer #1: Yes

Reviewer #2: Yes

3. Have the authors made all data underlying the findings in their manuscript fully available?

Reviewer #1: Yes

Reviewer #2: Yes

4. Is the manuscript presented in an intelligible fashion and written in standard English?

Reviewer #1: Yes

Reviewer #2: Yes

5. Review Comments to the Author

Reviewer #1: This study reports the development and validation of the UNESP-Botucatu composite pain scale for assessing postoperative pain in pigs. It includes an ethogram for the development of the scale and the validation based on video footage from 45 piglets undergoing castration. Four moments were assessed by 4 different observers. Validation of the scale and definition of the cut-off for administration of rescue analgesia were performed based on solid statistics. The study is well designed, conducted and reported.

Most of this reviewer’s comments were made for improving the readability and fluidity of the text as well as for providing more details and clarifying some points of the methodology and results sections.

Abstract:

Line 1: Perhaps change “mandatory” to “essential”?

Line 3: change to “treatments”

Line 5: change “submitted to” to “undergoing”

Lines 6-9: Suggest changing to: “Behavior was video-recorded 30 minutes before and intermittently up to 24 hours after castration. Edited footage (before surgery, after surgery before and after rescue analgesia, and 24 hours postoperatively) was analysed twice (one month apart) by one observer who was present during video-recording and three blinded observers”.

Line 13: Suggest changing to: “except between observers 1 and 3 (moderate agreement; 0.57)”

Line 14: Remove “the” from “based on the principal component”

Introduction:

The Introduction is very nice and it touches the most important points pertaining to the study. However, it is overly long. Please edit the introduction to shorten the amount of text.

Page 3, Line 7: Change to: “…respectively. However, among..”

Page 3, Lines 15-16: Please clarify that the data on beef and dairy cattle refer to animals undergoing the same procedure (castration?).

Pages 3 and 4, Lines 24-2: Suggest changing to: “This paradigm has been challenged by the fact that pigs treated with anaesthesia and analgesia showed increased short- and long-term weight gain when compared with pigs castrated without anaesthesia; thus, providing an economic benefit to producers.”

Page 4, Lines 6-8: Please split this sentence in two (one for Canada and another for USA and Europe).

Page 4, Line 21: Change “determination” for “assessment”

Page 5, Lines 4-13: Suggest deleting this paragraph. It does not add much relevant information.

Page 6, Line 4: Please correct: “this species” (singular)

Page 6, Lines 1-14: There is a link missing between these paragraphs. Please edit this section to shorten and link them.

Page 5, Lines 20-24 and Page 6, Lines 15-18: These two sections should be together.

Materials and methods

Page 7, Line 12: Was food also available ad libitum? Please clarify

Page 7, Line 14: Please remove the comma “..maternity period when piglets”

Page 7, Lines 14-16: Was the observer actively making physical contact with the piglets in the first week? Please clarify what you mean by “friendly contact with the piglets”. Also, I am not sure what you mean by “the observer followed the daily routine of the sector”.

Page 7, Line 22: 1 mL divided into two doses of 0.5 mL into each testicle? Please clarify.

Page 8, Line 8: When you say “evaluator”, I assume you refer to the observer who had been familiarizing with the pigs. It does not seem that this individual was actually ‘evaluating’ pain, correct?

Please clarify the presence of the observer. Was it only to set the camera? Was he/she the one evaluating the surgical wound and/or giving the medications?

Please also specify the sex of this individual and the other 3 observers.

Page 8, Lines 10-17: How long were the selected footage for each moment? I assume the individual who edited the footage was the same familiarizing with the pigs and doing the video recording?

Also, you might want to use “;” instead of “,” between moments and put explanations between parenthesis to facilitate the reading since this sentence has 7 lines. Consider removing the information being repeated in Figure 1.

Page 9 Line 5: Do you mean “behaviours observed in the pilot study?”. It seems like there is something missing here.

Page 9 Line 5: “were accepted”. Do you mean that they were included in the scale?

Page 9 Lines 13-16: What do you mean by “variable counting scale”?

Page 9 Lines 17-20: Please remove. You already partially said this. No need to repeat. Only clarify on page 8 lines 10-17 that the edited videos were each 4 minutes long.

Page 11 Lines 12-13: Description of the Altman’s classification should be reported in line 3 since this is the first time you refer to it.

Results

Table 1: I don’t see the explanation for the Asterix. Please include it.

The behavior “rubbing the affected area on the floor” is described under ‘Altered posture’ and ‘Lie down uncomfortably’. How did the observer choose to classify this behavior since it is described under two different categories?

Table 2: “percentage presence”? Please reword.

Please explain the superscript letters.

Page 18 Line 5: Correct “Table 2”. Not “Table 1”.

Table 3: Excellent idea to have the videos in the table! Please recheck the links as some videos are not working.

Table 4: You already explained in the title that values are reported as ICC (CI), so perhaps you can consider deleting the second line of the table [ICC (CI].

If the ICC used for evaluating criterion validity is the ICC between the gold standard and the other observers, how come the ICC between gold standard and gold standard is 0.88? I would expect it to be 1. The same would be for ICC between Observer 1 and Observer 1, and so on. It might be that I don’t understand the statistics of this, but regardless, it’s probably a good idea to explain it a bit more.

Criterion validity would only consider the ICC between the gold standard and each of the other observers, correct? In that case, do you need to report the ICC between observers 1, 2 and 3?

Table 8: Please add the explanation of the Asterix in the table legend.

Perhaps it would be nice to also explain what these ‘dimensions’ are in the legend.

Page 26, Line 1: Please remove ‘the’ from “analysis at the four”.

Table 11: In Page 10, Lines 6-7, it is not clear that the scores used for definition of the cut-off for rescue analgesia were only the scores from the second phase of video analysis. Please clarify.

Page 32 Lines 9-11. This sentence is confusing. Please reword.

Discussion

Page 37 Line 4: Correct “this species”

Page 37 Line 22: Remove “only”

Page 38 Line 5: Consider changing “tool is carried out” for “tool should be carried out”

Page 38 Lines 17-20: This sentence is confusing. Please rephrase.

Page 38 Lines 22-24: Do you mean that the observers were selected because they had experience in pain scoring in cattle? This section is not clear. Indeed, detailed information on the sex and level of training of observers is not clear in the M&M. I even wonder if you need to discuss them here or if clearly explaining them in the M&M is enough. I’ll leave this to your discretion.

Page 39 Lines 4-8: You used this as an additional means of assessing construct validity, correct? In that case, this paragraph should be linked to Page 38 Lines 3-11. It seems a bit lost herein.

Page 39 Lines 16-17: I don’t see a link for the sentence “Therefore, … factor analysis”. Please reword or delete this sentence.

Page 41 Lines 6-9: In this sentence you refer to production and research animals. But the sentence is constructed in a manner that only refers to production animals. Please reword.

Page 42 Line 2: Suggest changing to: “pain-related behavior changes were more evident”

Page 42 Lines 6-12: Perhaps add a small sentence related to the advantage of video analysis as the presence of the observer might affect pain-related behaviors?

Reviewer #2: Dear Authors,

thank you very much for your paper. it covers a very useful topic and is very well conducted, despite the huge amount of data analyzed and discussed.

I have only a few suggestions, that I attach below:

INTRODUCTION

Lines 15-16 (pag. 5): I would eliminate the sentence: Providing that a pain-related behaviour scale is validated, the instrument should present reliability, specificity, and sensitivity.

Line 24 (pag. 5): for cats, I would add the reference by Reid et al.: Definitive Glasgow acute pain scale for cats - validation and intervention level

METHODS

Lines 24 (pag 6)-2 (pag 7): I would move the sentence: “A pilot study was carried out with 12 piglets aged 35 days, submitted to castration to define the experimental protocol. For the main study, 45 Landrace, Large White, Duroc, and Hampshire male pigs were randomly selected. The pigs were aged 1 38±3 days (range2 35 – 41 days) and weighed 11.06 ± 2.28 kg” after the inclusion and exclusion criteria at the end of line 8 (pag. 7)

Line 17 (pag. 8): move here, after “(M4)”, the citation to fig. 1

Line 19: eliminate citation to fig. 1 here

Line 8 (page 10): the sentence should be changed as follow: Subsequently, they were required to ascribe a score for the developed scale as well as the visual analogue scale (VAS), numerical rating scale (NRS), where….

STATISTICAL ANALYSIS

Line 21-22 (pag. 10): As here you are talking about inter-observer reliability and not about criterion validity, I would change the sentence: “comparing the gold-standard assessments against the other observers respectively” in “comparing the degree of agreement among different observers”.

Lines 12-13 (pag. 11): I would delete here the specifications about the Altman’s classification and move it at the end of line 3.

Lines 18-20 (page 11): Is this sentence correct?

Line 2 (pag 13): delete “S” before “< 70%”

RESULTS:

Tab 1: What “*” means?

Tab. 2: the legend should read: “Median and amplitude of the presence (number of minutes of each behaviour/30 minutes) of normal behaviours,….

Line 5 (pag. 18): eliminate the citation to Table 1 here. At the end of the paragraph (line 10) you should cite Table 2.

Line 1-11 (pag 21): the title and the paragraph, as well as table 4, is a bit confusing: in the title you put altogether intra- and inter-observer reliability and criterion validity, in the text you mix criterion validity (first part of the sentence (lines 3-4) with inter-observer reliability (lines 4-5) and you do not mention at all intra-observer reliability.

Later on, you have dedicated paragraphs on Intra-observer reliability and Criterion validity (see my comments to line 9 of pag 22).

Line 9 (pag. 22) the title of the paragraph should read “Criterion validity”!

Line 1 (pag. 23): the Title of table 6 should refer to criterion validity and not to inter-observer reliability

Line 10 (pag. 23): I Would keep in the title only “Concurrent validity”

Line 12 (pag. 25): fig. 2 is not cited in the text. Maybe you can cite it together with Table 8 (pag. 24)

Line 12 (pag. 27): move “(table 10)” at the end of the following sentence (line 14).

Line 19 (pag. 32): fig. 4 is not cited in the text. Maybe you can cite it together with Fig. 3 (line 11)

Line 4 (pag. 33): it should read “Fig 5”!?!

DISCUSSION

Line 8 (pag. 38): I would refer to “construct validity” and not to concurrent validity as you are referring to the comparison with the “gold standard” and not with VAS, NAS and SDS.

Line 5 (pag. 39): in order to avoid confusion, I would use the term “concurrent validity” instead of “reliability”.

6. PLOS authors have the option to publish the peer review history of their article (what does this mean?). If published, this will include your full peer review and any attached files.

Reviewer #1: No

Reviewer #2: Yes: Giorgia della Rocca

---

## [Author Response · Author response to Decision Letter 0]

6 Mar 2020

Additional Editor Comments (if provided):

Dear Authors,

it seems that both the reviewer have some minor related to your study. Please provide a complete response or amend the manuscript accordingly their suggestions if possible

The authors appreciate the reviewers for taking the time to review this manuscript carefully. 

In the meantime the manuscript was reviewed, the authors developed a further analysis of data. We would appreciate the permission of the editor to include data regarding the predictive criterion validity (Table 8), the Youden indexes of the unidimensional scales (Table 16), the percentage of pigs present in the diagnostic uncertainty zone (Table 17) and finally, the most essential data in our opinion, the classification of pain intensity according non-hierarchical cluster analysis (Fig 6 and 7). The authors believe these inclusions will strengthen the validation process. 

To compensate these inclusions, the Introduction section and Table 15 were reduced and individual data from Fig 5 were grouped in one graph instead of 4.

All corrections were performed as suggested. Please see the comments below.

Journal Requirements:

Answer: these links do not seem to work, but we followed instructions for authors

2. In your Methods section, please provide additional details regarding the animals used in your study and ensure you have described the source. For more information regarding PLOS' policy on materials sharing and reporting, see https://journals.plos.org/plosone/s/materials-and-software-sharing#loc-sharing-materials.

Answer: This information was included (page 6, line 7)

3. In your Methods section, please include a comment about the state of the animals following this research. Were they euthanized or housed for use in further research? If any animals were sacrificed by the authors, please include the method of euthanasia and describe any efforts that were undertaken to reduce animal suffering.

Answer: This information was included (page 7 – line 14-17). After the end of the field study pigs were kept until 145 days of age when they were submitted to humane slaughter accomplished by electrical stunning, followed by exsanguination, according to the Brazilian Federal legislation (Ministry of Agriculture).

4. Please amend the manuscript submission data (via Edit Submission) to include author Flávia Augusta de Oliveira.

Answer: Flavia Oliveira was included

5. Please ensure that you refer to Figure 2 and 5 in your text as, if accepted, production will need this reference to link the reader to the figure.

Answer: corrected

6. Please upload a new copy of Figure 5 as the detail is not clear. Please follow the link for more information: http://blogs.PLOS.org/everyone/2011/05/10/how-to-check-your-manuscript-image-quality-in-editorial-manager/

Answer: All figures were uploaded after using PACE

7. We note you have included a table to which you do not refer in the text of your manuscript. Please ensure that you refer to Table 2 in your text; if accepted, production will need this reference to link the reader to the Table.

Answer: included

Answer: included

Reviewers' comments:

Reviewer's Responses to Questions

Comments to the Author

1. Is the manuscript technically sound, and do the data support the conclusions?

Reviewer #1: Yes

Reviewer #2: Yes

 2. Has the statistical analysis been performed appropriately and rigorously?

Reviewer #1: Yes

Reviewer #2: Yes

3. Have the authors made all data underlying the findings in their manuscript fully available?

Reviewer #1: Yes

Reviewer #2: Yes

 4. Is the manuscript presented in an intelligible fashion and written in standard English?

Reviewer #1: Yes

Reviewer #2: Yes

 5. Review Comments to the Author

Reviewer #1: This study reports the development and validation of the UNESP-Botucatu composite pain scale for assessing postoperative pain in pigs. It includes an ethogram for the development of the scale and the validation based on video footage from 45 piglets undergoing castration. Four moments were assessed by 4 different observers. Validation of the scale and definition of the cut-off for administration of rescue analgesia were performed based on solid statistics. The study is well designed, conducted and reported.

Most of this reviewer’s comments were made for improving the readability and fluidity of the text as well as for providing more details and clarifying some points of the methodology and results sections.

The authors appreciate the reviewer for taking the time to review this manuscript carefully. 

In the meantime the manuscript was reviewed, the authors developed a further analysis of data. We would appreciate the permission of the reviewer to include data regarding the predictive criterion validity (Table 8), the Youden indexes of the unidimensional scales (Table 16), the percentage of pigs present in the diagnostic uncertainty zone (Table 17) and finally, the most essential data in our opinion, the classification of pain intensity according non-hierarchical cluster analysis (Fig 6 and 7). The authors believe these inclusions will strengthen the validation process. 

To compensate these inclusions, the Introduction section and Table 15 were reduced and individual data from Fig 5 were grouped in one graph instead of 4.

All corrections were performed as suggested. Please see the comments below.

Abstract:

Line 1: Perhaps change “mandatory” to “essential”?

Line 3: change to “treatments”

Line 5: change “submitted to” to “undergoing”

Lines 6-9: Suggest changing to: “Behavior was video-recorded 30 minutes before and intermittently up to 24 hours after castration. Edited footage (before surgery, after surgery before and after rescue analgesia, and 24 hours postoperatively) was analysed twice (one month apart) by one observer who was present during video-recording and three blinded observers”.

Line 13: Suggest changing to: “except between observers 1 and 3 (moderate agreement; 0.57)”

Line 14: Remove “the” from “based on the principal component”

Answer: All corrections were performed.

Introduction:

The Introduction is very nice and it touches the most important points pertaining to the study. However, it is overly long. Please edit the introduction to shorten the amount of text.

Answer: Introduction section was reduced of about 25%.

Page 3, Line 7: Change to: “…respectively. However, among..”

Page 3, Lines 15-16: Please clarify that the data on beef and dairy cattle refer to animals undergoing the same procedure (castration?).

Pages 3 and 4, Lines 24-2: Suggest changing to: “This paradigm has been challenged by the fact that pigs treated with anaesthesia and analgesia showed increased short- and long-term weight gain when compared with pigs castrated without anaesthesia; thus, providing an economic benefit to producers.”

Page 4, Lines 6-8: Please split this sentence in two (one for Canada and another for USA and Europe).

Page 4, Line 21: Change “determination” for “assessment”

Page 5, Lines 4-13: Suggest deleting this paragraph. It does not add much relevant information.

Page 6, Line 4: Please correct: “this species” (singular)

Page 6, Lines 1-14: There is a link missing between these paragraphs. Please edit this section to shorten and link them.

Page 5, Lines 20-24 and Page 6, Lines 15-18: These two sections should be together.

Answer: All corrections were performed and introduction was condensed.

Materials and methods

Page 7, Line 12: Was food also available ad libitum? Please clarify

Answer: Information was included. They were fed three times a day and food was also offered during the perioperative pain assessment moments (page 6, line 15; oage 7, line 12-14)).

Page 7, Line 14: Please remove the comma “..maternity period when piglets”

Page 7, Lines 14-16: Was the observer actively making physical contact with the piglets in the first week? Please clarify what you mean by “friendly contact with the piglets”. Also, I am not sure what you mean by “the observer followed the daily routine of the sector”.

Answer: more detail was included to make things clearer. “After a week of contact with the piglets, by cleaning the pen and providing food, the in-person researcher initiated direct friendly contact inside the stall, for 15 minutes three times a day, without making any sudden movement, or using vocal communication, so that the piglets could spontaneously approach” (page 6, line 18-21. The pigs by this stage were playing with the in-person researcher, by smelling, nibbling and pulling clothes and hair.

Page 7, Line 22: 1 mL divided into two doses of 0.5 mL into each testicle? Please clarify.

Answer: This was clarified. Pigs received 0.5 mL of 1% lidocaine at each incision line, and 1 mL injected intratesticularly at each testicle (page 6, line 24-25 and page 7, line 1).

Page 8, Line 8: When you say “evaluator”, I assume you refer to the observer who had been familiarizing with the pigs. It does not seem that this individual was actually ‘evaluating’ pain, correct?

Answer: The local (in-person) observer was replaced by in-person researcher to avoid misinterpretation throughout the manuscript.

Please clarify the presence of the observer. Was it only to set the camera? Was he/she the one evaluating the surgical wound and/or giving the medications?

Answer: The local (in-person) observer was replaced by in-person researcher to avoid misinterpretation throughout the manuscript. 

Please also specify the sex of this individual and the other 3 observers.

Answer: included. One man and three women.

Page 8, Lines 10-17: How long were the selected footage for each moment? I assume the individual who edited the footage was the same familiarizing with the pigs and doing the video recording?

Answer: This was included in line 10-11 (Page 7). This information was included again in line 17, page 8.

Also, you might want to use “;” instead of “,” between moments and put explanations between parenthesis to facilitate the reading since this sentence has 7 lines. Consider removing the information being repeated in Figure 1.

Answer: This sentence was removed and all information included in Figure 1 as suggested.

Page 9 Line 5: Do you mean “behaviours observed in the pilot study?”. It seems like there is something missing here.

Answer: “The scale was developed according to the analysis of the relevance of pain-related behaviours previously described in the literature[6,8,32-35] and behaviours observed both in the pilot study and during video edition by the in-person researcher.” (page 8, line 4-6)

Page 9 Line 5: “were accepted”. Do you mean that they were included in the scale?

Answer: Yes. Corrected

Page 9 Lines 13-16: What do you mean by “variable counting scale”?

Answer: excluded because this is irrelevant and to avoid misinterpretation.

Page 9 Lines 17-20: Please remove. You already partially said this. No need to repeat. Only clarify on page 8 lines 10-17 that the edited videos were each 4 minutes long.

Answer: partially deleted (page 8, line 17-18)

Page 11 Lines 12-13: Description of the Altman’s classification should be reported in line 3 since this is the first time you refer to it.

Answer: corrected

Results

Table 1: I don’t see the explanation for the Asterix. Please include it.

Answer: asterisks were deleted as they mean nothing

The behavior “rubbing the affected area on the floor” is described under ‘Altered posture’ and ‘Lie down uncomfortably’. How did the observer choose to classify this behavior since it is described under two different categories?

Answer: Please accept our apologies for this mistake. This behaviour was classified as ‘altered posture’ when pigs rubbed the affected area against any structure (like the grid) only when standing; therefore ‘on the floor’ was deleted from here. Otherwise, when pigs rubbed the affected area specifically on the floor when they were lying down, this behaviour was included in the item ‘Lie down uncomfortably’. For the item ‘Attention to the affected area’ of the scale, the behaviour ‘scratches/rubs the painful area’ was scored both when pigs scratched on the floor or against any surface, like the grid.

Table 2: “percentage presence”? Please reword.

Answer: corrected

Please explain the superscript letters.

Answer: included

Page 18 Line 5: Correct “Table 2”. Not “Table 1”.

Table 3: Excellent idea to have the videos in the table! Please recheck the links as some videos are not working.

Answer: The authors acknowledge your comment. We rechecked again and they are all working in Google Chrome and Internet explorer. If the reviewer copy and paste straight to the browser they should work. Could the reviewer please be kind enough to let us know which link is not working? 

Table 4: You already explained in the title that values are reported as ICC (CI), so perhaps you can consider deleting the second line of the table [ICC (CI].

Answer: deleted

If the ICC used for evaluating criterion validity is the ICC between the gold standard and the other observers, how come the ICC between gold standard and gold standard is 0.88? I would expect it to be 1. The same would be for ICC between Observer 1 and Observer 1, and so on. It might be that I don’t understand the statistics of this, but regardless, it’s probably a good idea to explain it a bit more.

Answer: Please accept our apologies. We double checked the data. The inter-observer data were correct, but the intra-observer was not. This was amended and duplicate data were removed to facilitate reading (now this is Table 6 as Criterion validity data were placed together, according to Reviewer 2 suggestion)

Criterion validity would only consider the ICC between the gold standard and each of the other observers, correct? In that case, do you need to report the ICC between observers 1, 2 and 3?

Answer: If the reviewer agrees, the authors think it is interesting to inform the correlation among all observers for comparison as well, however as mentioned before duplicate data were removed to facilitate reading (now this is Table 6 as Criterion validity data were placed together, according to Reviewer 2 suggestion).

Table 8: Please add the explanation of the Asterix in the table legend.

Perhaps it would be nice to also explain what these ‘dimensions’ are in the legend.

Answer: Now table 9. Asterisk was included in the legend, and so was additional information regarding this analysis. Please see the explanation of the legend of Figure 2 as well.

Page 26, Line 1: Please remove ‘the’ from “analysis at the four”.

Answer: corrected

Table 11: In Page 10, Lines 6-7, it is not clear that the scores used for definition of the cut-off for rescue analgesia were only the scores from the second phase of video analysis. Please clarify.

Answer: The authors presume the reviewer is referring to Table 14 (now Table 15)? This information was included. Only one ROC curve by grouping all observers replaced the individual ROC curves.

Page 32 Lines 9-11. This sentence is confusing. Please reword.

Answer: Amended (page 30, line 4-11).

Discussion

Page 37 Line 4: Correct “this species”

Page 37 Line 22: Remove “only”

Page 38 Line 5: Consider changing “tool is carried out” for “tool should be carried out”

Answer: Amended

Page 38 Lines 17-20: This sentence is confusing. Please rephrase.

Answer: The sentence was excluded, and the paragraph was shortened. Content validation was not only based on “expert” opinion, but also included the ethogram findings and behaviour described in the literature (page 36, line 10-14).

Page 38 Lines 22-24: Do you mean that the observers were selected because they had experience in pain scoring in cattle? This section is not clear. Indeed, detailed information on the sex and level of training of observers is not clear in the M&M. I even wonder if you need to discuss them here or if clearly explaining them in the M&M is enough. I’ll leave this to your discretion.

Answer: More detailed information regarding the observers were include in Methods section (page 9, line 1-6) and the paragraph was rephrased (page 36, line 2-5).

Page 39 Lines 4-8: You used this as an additional means of assessing construct validity, correct? In that case, this paragraph should be linked to Page 38 Lines 3-11. It seems a bit lost herein.

Answer: Thanks for pointing that. This part is related to criterion validity. These sentences were placed together and followed the order of results to maintain consistency (page 36, line 15-24, page 37, line 1-2.

Page 39 Lines 16-17: I don’t see a link for the sentence “Therefore, … factor analysis”. Please reword or delete this sentence.

Answer: deleted

Page 41 Lines 6-9: In this sentence you refer to production and research animals. But the sentence is constructed in a manner that only refers to production animals. Please reword.

Answer: the paragraph was shortened and rephrased (page 39, line 1-7).

Page 42 Line 2: Suggest changing to: “pain-related behavior changes were more evident”

Answer: corrected (page 40, line 7)

Page 42 Lines 6-12: Perhaps add a small sentence related to the advantage of video analysis as the presence of the observer might affect pain-related behaviors?

Answer: included (page 40, line 10-18)

Reviewer #2: Dear Authors,

thank you very much for your paper. it covers a very useful topic and is very well conducted, despite the huge amount of data analyzed and discussed.

I have only a few suggestions, that I attach below:

The authors appreciate the reviewer for taking the time to review this manuscript carefully. 

In the meantime the manuscript was reviewed, the authors developed a further analysis of data. We would appreciate the permission of the reviewer to include data regarding the predictive criterion validity (Table 8), the Youden indexes of the unidimensional scales (Table 16), the percentage of pigs present in the diagnostic uncertainty zone (Table 17) and finally, the most essential data in our opinion, the classification of pain intensity according non-hierarchical cluster analysis (Fig 6 and 7). The authors believe these inclusions will strengthen the validation process. 

To compensate these inclusions, the Introduction section and Table 15 were reduced and individual data from Fig 5 were grouped in one graph instead of 4.

All corrections were performed as suggested. Please see the comments below.

INTRODUCTION

Lines 15-16 (pag. 5): I would eliminate the sentence: Providing that a pain-related behaviour scale is validated, the instrument should present reliability, specificity, and sensitivity.

Answer: deleted as requested

Line 24 (pag. 5): for cats, I would add the reference by Reid et al.: Definitive Glasgow acute pain scale for cats - validation and intervention level

Answer: included

METHODS

Lines 24 (pag 6)-2 (pag 7): I would move the sentence: “A pilot study was carried out with 12 piglets aged 35 days, submitted to castration to define the experimental protocol. For the main study, 45 Landrace, Large White, Duroc, and Hampshire male pigs were randomly selected. The pigs were aged 1 38±3 days (range2 35 – 41 days) and weighed 11.06 ± 2.28 kg” after the inclusion and exclusion criteria at the end of line 8 (pag. 7)

Answer: changed as requested (page 6, line 7-11).

Line 17 (pag. 8): move here, after “(M4)”, the citation to fig. 1

Answer: The description of moments were maintained only in Fig 1 as requested by Reviewer 1. The citation of Fig 1 was inserted in line 20, page 7.

Line 19: eliminate citation to fig. 1 here

Answer: excluded

Line 8 (page 10): the sentence should be changed as follow: Subsequently, they were required to ascribe a score for the developed scale as well as the visual analogue scale (VAS), numerical rating scale (NRS), where….

Answer: We apologize but it is necessary to maintain as before as the observers first scored the unidimensional scales and then the proposed scale. We condensed the explanation about the scores and placed between brackets (page 9, line 8-14). 

STATISTICAL ANALYSIS

Line 21-22 (pag. 10): As here you are talking about inter-observer reliability and not about criterion validity, I would change the sentence: “comparing the gold-standard assessments against the other observers respectively” in “comparing the degree of agreement among different observers”.

Answer: amended (page 9, line 22-23)

Lines 12-13 (pag. 11): I would delete here the specifications about the Altman’s classification and move it at the end of line 3.

Answer: corrected

Lines 18-20 (page 11): Is this sentence correct?

Answer: Corrected to “The factorial structure was confirmed when items showed a factor load ≥ 0.50 or ≤ -0.50” (page 10, line 24-25).

Line 2 (pag 13): delete “S” before “< 70%”

Answer: corrected

RESULTS:

Tab 1: What “*” means?

Answer: asterisks were deleted as they mean nothing

Tab. 2: the legend should read: “Median and amplitude of the presence (number of minutes of each behaviour/30 minutes) of normal behaviours,….

Answer: corrected

Line 5 (pag. 18): eliminate the citation to Table 1 here. At the end of the paragraph (line 10) you should cite Table 2.

Answer: corrected

Line 1-11 (pag 21): the title and the paragraph, as well as table 4, is a bit confusing: in the title you put altogether intra- and inter-observer reliability and criterion validity, in the text you mix criterion validity (first part of the sentence (lines 3-4) with inter-observer reliability (lines 4-5) and you do not mention at all intra-observer reliability.

Later on, you have dedicated paragraphs on Intra-observer reliability and Criterion validity (see my comments to line 9 of pag 22).

Answer: Thanks for pointing that. Table 4 was changed to Table 6 and positioned in Criterion validity section to keep consistency.

Line 9 (pag. 22) the title of the paragraph should read “Criterion validity”!

Answer: corrected as described above. Matrix correlation is Criterion validity

Line 1 (pag. 23): the Title of table 6 should refer to criterion validity and not to inter-observer reliability

Answer: Table 6 (now Table 5) is not Criterion validity, therefore table 4 (now Tale 6 - Matrix coefficient) was positioned in Criterion validity section to maintain coherence, as previously described.

Line 10 (pag. 23): I Would keep in the title only “Concurrent validity”

Answer: changed to Criterion validity because a new data were incorporated, therefore, criterion validity was subdivided in concurrent and predictive criterion validity page 21, line 6).

Line 12 (pag. 25): fig. 2 is not cited in the text. Maybe you can cite it together with Table 8 (pag. 24)

Answer: included

Line 12 (pag. 27): move “(table 10)” at the end of the following sentence (line 14).

Answer: replaced

Line 19 (pag. 32): fig. 4 is not cited in the text. Maybe you can cite it together with Fig. 3 (line 11)

Answer: Fig. 3 and 4 were cited in the first sentence (page 30, line 4)

Line 4 (pag. 33): it should read “Fig 5”!?!

Answer: corrected

DISCUSSION

Line 8 (pag. 38): I would refer to “construct validity” and not to concurrent validity as you are referring to the comparison with the “gold standard” and not with VAS, NAS and SDS.

Answer: In this study there were two approaches to perform concurrent criterion validity. One was by comparing the scale against the unidimensional scales and the other one was to compare the gold standard assessment against the other observers. Construct validity is related to responsiveness (Streiner DL, Norman GR, Cairney J. Health measurement scales: a practical guide to their development and use. 5th ed. Oxford: Oxford University Press; 2015)

Line 5 (pag. 39): in order to avoid confusion, I would use the term “concurrent validity” instead of “reliability”.

Answer: Thanks for your suggestion. This was corrected (page 36, line 23).

---

## [Decision Letter · Decision Letter 1]

2 Apr 2020

PONE-D-19-30670R1

Validation of the UNESP-Botucatu pig composite acute pain scale (UPAPS)

PLOS ONE

Dear Prof Stelio Pacca Loureiro Luna

Thank you for submitting your manuscript to PLOS ONE. After careful consideration, we feel that it has merit but does not fully meet PLOS ONE’s publication criteria as it currently stands. Therefore, we invite you to submit a revised version of the manuscript that addresses the points raised during the review process.

Dear Authors,

please provide an adequate answer to the minor critiques of the first reviewer.

We would appreciate receiving your revised manuscript by May 17 2020 11:59PM. To enhance the reproducibility of your results, we recommend that if applicable you deposit your laboratory protocols in protocols.io, where a protocol can be assigned its own identifier (DOI) such that it can be cited independently in the future. For instructions see: http://journals.plos.org/plosone/s/submission-guidelines#loc-laboratory-protocols

We look forward to receiving your revised manuscript.

Kind regards,

Francesco Staffieri

Academic Editor

PLOS ONE

Additional Editor Comments (if provided):

Dear Authors,

please provide an adequate answer to the minor critiques of the first reviewer.

Reviewers' comments:

Reviewer's Responses to Questions

**Comments to the Author**

1. If the authors have adequately addressed your comments raised in a previous round of review and you feel that this manuscript is now acceptable for publication, you may indicate that here to bypass the “Comments to the Author” section, enter your conflict of interest statement in the “Confidential to Editor” section, and submit your "Accept" recommendation.

Reviewer #1: (No Response)

Reviewer #2: All comments have been addressed

2. Is the manuscript technically sound, and do the data support the conclusions?

Reviewer #1: Yes

Reviewer #2: Yes

3. Has the statistical analysis been performed appropriately and rigorously? 

Reviewer #1: Yes

Reviewer #2: Yes

4. Have the authors made all data underlying the findings in their manuscript fully available?

Reviewer #1: Yes

Reviewer #2: Yes

5. Is the manuscript presented in an intelligible fashion and written in standard English?

Reviewer #1: Yes

Reviewer #2: Yes

6. Review Comments to the Author

Reviewer #1: Thank you for addressing the concerns raised in the first review. The manuscript has much improved and there are only minor comments now. The main thing I think is still missing is a clear description of what constituted the pilot study and what constituted the validation study. The authors are congratulated on advancing pig welfare with this research.

Page 2 Line 12. Please removed ‘of’ from: “most of observers”

Page 2 Lines 18-19: Please reword: “increase and decrease in pain scores of all items of the scale after castration and intervention analgesia, respectively, according to..)

Page 6 Line 24: I would assume you need to specific the manufacturer for the lidocaine. Please check the Journal’s instructions. Same for the other drugs.

Page 7 Lines 18-19 Is this the pilot study? Please clarify

Page 7 Lines 21-22: Replace “time in seconds spent on” by “duration”

Fig 1 legend: Reword: “..moments used for validation”

Page 8 Line 17 Please reword “The in-person researcher edited each of the four 30-minute videos into 4-minute videos representing the 4 moments..”

Page 9 Line 3: Change to ‘Developed”

Page 10 Line 12: “confidence interval” – You already defined this abbreviation on line 1 of the same page.

Page 13 Lines 10-13: This is still a bit confusing, and I think it is because it is not entirely clear from M&M what was the pilot study for development of the ethogram and what was the validation study. Please make sure this is clear in the M&M and in the results. Consider rewording this sentence.

Page 16 Line 7-9: The words “after surgery” should be at the end of the sentence in line 9.

Page 16 Line 13-14: In Page 8 line 13, you refer to ‘six behavioral categories’, here you refer to ‘items’. Please standardize the wording.

Table 3. Posture (3); Interaction (0, 1,2,3); Activity (0, 2, 3); Appetite (0,1,2,3); Attention (B, C,D); Miscellaneous (A, B, C, D): the link does not work, but copying and pasting the web address works.

Table 13 – Is this table really necessary in the manuscript? Perhaps as supplementary material?

Page 36 Line 4: ‘before the observations’

Page 37 Line 1: You are repeating the same information from line 18 in the previous page

Reviewer #2: Dear Authors, thanks so much for your revision. Now the paper sounds really good!!

I only have remained 2 small observations, but it's up to you if make the changes or not (in the case I have misunderstood).

1) In M&M you report that video-observers are asked to state if they would provide rescue analgesia and to ascribe a score for VAS, NAS and SDS, but you don't report that they also had to score the UPAPS!!.

So, I would read line 9-12 (page 9) as: "they were required to ascribe a score for the visual analogue scale (VAS), numerical rating scale (NRS; 0 “no pain” to 10 “worst possible pain”), and the simple descriptive scale (SDS; 0 - no pain to 3 - intense pain), AND THEN TO SCORE THE PROPOSED SCALE.

2) In my experience, Table 5 shows results for Concurrent criterion validity (i.e. comparison between the gold standard and each observer), and not for inter-observer reliability that, as stated in lines 22-23 (page 9) refers to the degree of agreement among observers. It's true that also the agreement between the gold standard and each single observer can be part of the inter-observer reliability, but in this table data showing inter-observer reliability (comparison among the 3 observers) are non shown... Indeed, in the first line of table 6 (that is cited in the text as showing data for criterion validity), you repeat results shown in table 5... And I'm still not able to understand the following 2 lines of table 6... are you comparing here data among observers? And if yes, why Observer 2 is not matched with Observer 1?

Still a bit confusing...

7. PLOS authors have the option to publish the peer review history of their article (what does this mean?). If published, this will include your full peer review and any attached files.

Reviewer #1: No

Reviewer #2: Yes: Giorgia della Rocca

---

## [Author Response · Author response to Decision Letter 1]

8 Apr 2020

Answers to reviewers

Reviewers' comments:

Reviewer #1: Thank you for addressing the concerns raised in the first review. The manuscript has much improved and there are only minor comments now. The main thing I think is still missing is a clear description of what constituted the pilot study and what constituted the validation study. The authors are congratulated on advancing pig welfare with this research.

Answer: the authors are very grateful for the suggestions which were fully incorporated.

Page 2 Line 12. Please removed ‘of’ from: “most of observers”

Answer: corrected (page 2, line 13)

Page 2 Lines 18-19: Please reword: “increase and decrease in pain scores of all items of the scale after castration and intervention analgesia, respectively, according to..)

Answer: corrected and rephrased (page 2, line 18-19)

Page 6 Line 24: I would assume you need to specific the manufacturer for the lidocaine. Please check the Journal’s instructions. Same for the other drugs.

Answer: included (page 7, line 6-7, 12, 15-16)

Page 7 Lines 18-19 Is this the pilot study? Please clarify

Answer: “from the main study” was included (page 8, line 3)

Page 7 Lines 21-22: Replace “time in seconds spent on” by “duration”

Answer: corrected (page 8, line 7)

Fig 1 legend: Reword: “..moments used for validation”

Answer: corrected to “Timeline of interventions, drugs and pain assessments performed for validation of the Unesp-Botucatu pig composite acute pain scale (UPAPS)” (page 8; line 9)

Page 8 Line 17 Please reword “The in-person researcher edited each of the four 30-minute videos into 4-minute videos representing the 4 moments..”

Answer: corrected (page 9, line 1-2)

Page 9 Line 3: Change to ‘Developed”

Answer: corrected (page 9, line 5)

Page 10 Line 12: “confidence interval” – You already defined this abbreviation on line 1 of the same page.

Answer: ‘confidence interval’ was excluded from here

Page 13 Lines 10-13: This is still a bit confusing, and I think it is because it is not entirely clear from M&M what was the pilot study for development of the ethogram and what was the validation study. Please make sure this is clear in the M&M and in the results. Consider rewording this sentence.

Answer: “A pilot study was carried out with 12 piglets aged 35 days, submitted to castration only to define the experimental protocol and to recognize pain-related behaviours.” (page 6; line 15-16); 

“Footage from 40 out of 45 pigs was used for the ethogram because in 5 pigs footages were recorded for slightly less than 30 minutes at some moments; therefore data from these 5 five pigs were excluded from the ethogram analysis. For validation of the scale, 180 videos of approximately 4-minute durations were edited and footage of the main study from 45 pigs were used” (page 13; line 17-21). 

Page 16 Line 7-9: The words “after surgery” should be at the end of the sentence in line 9.

Answer: corrected (page 16, line 9)

Page 16 Line 13-14: In Page 8 line 13, you refer to ‘six behavioral categories’, here you refer to ‘items’. Please standardize the wording.

Answer: ‘categories’ was replaced by ‘items’ in all cases

Table 3. Posture (3); Interaction (0, 1,2,3); Activity (0, 2, 3); Appetite (0,1,2,3); Attention (B, C,D); Miscellaneous (A, B, C, D): the link does not work, but copying and pasting the web address works.

Answer: all links are working correctly now without the need to copy and paste. We had to press ‘enter’ after the link to work out.

Table 13 – Is this table really necessary in the manuscript? Perhaps as supplementary material?

Answer: Table 13 was excluded from the main text and included as supplementary material (S1 Table) as required (page 28, line 9)

Page 36 Line 4: ‘before the observations’

Answer: corrected (page 37, line 5)

Page 37 Line 1: You are repeating the same information from line 18 in the previous page

Answer: “and this methodology has already been used in other studies in cats [21], cattle [22], and horses[27]” was excluded

 

Reviewer #2: Dear Authors, thanks so much for your revision. Now the paper sounds really good!!

I only have remained 2 small observations, but it's up to you if make the changes or not (in the case I have misunderstood).

The authors are very grateful for the previous and present suggestions which were fully incorporated.

1) In M&M you report that video-observers are asked to state if they would provide rescue analgesia and to ascribe a score for VAS, NAS and SDS, but you don't report that they also had to score the UPAPS!!.

So, I would read line 9-12 (page 9) as: "they were required to ascribe a score for the visual analogue scale (VAS), numerical rating scale (NRS; 0 “no pain” to 10 “worst possible pain”), and the simple descriptive scale (SDS; 0 - no pain to 3 - intense pain), AND THEN TO SCORE THE PROPOSED SCALE.

Answer: thanks for addressing that. This was included (page 9, line 14).

2) In my experience, Table 5 shows results for Concurrent criterion validity (i.e. comparison between the gold standard and each observer), and not for inter-observer reliability that, as stated in lines 22-23 (page 9) refers to the degree of agreement among observers. It's true that also the agreement between the gold standard and each single observer can be part of the inter-observer reliability, but in this table data showing inter-observer reliability (comparison among the 3 observers) are non shown... Indeed, in the first line of table 6 (that is cited in the text as showing data for criterion validity), you repeat results shown in table 5... And I'm still not able to understand the following 2 lines of table 6... are you comparing here data among observers? And if yes, why Observer 2 is not matched with Observer 1?

Still a bit confusing...

Answer: Although both Tables 5 and 6 report results of inter-observer reliability, the authors agree that for the concurrent criterion validity, results from the Gold standard should be compared to the other observers; therefore Tables 5 and 6 were swapped to consider this suggestion. We hope it is clearer now.

Table 5 (previous Table 6) expresses agreements among all observers. The comparison between Observer 1 and 2 is in the intersection between the 3rd line and 3rd column of table 5 (previous table 6).

---

## [Decision Letter · Decision Letter 2]

8 May 2020

Validation of the UNESP-Botucatu pig composite acute pain scale (UPAPS)

PONE-D-19-30670R2

Dear Dr. Stelio Pacca Loureiro Luna,

We are pleased to inform you that your manuscript has been judged scientifically suitable for publication and will be formally accepted for publication once it complies with all outstanding technical requirements.

With kind regards,

Francesco Staffieri

Academic Editor

PLOS ONE

Additional Editor Comments (optional):

Reviewers' comments:

Reviewer's Responses to Questions

**Comments to the Author**

1. If the authors have adequately addressed your comments raised in a previous round of review and you feel that this manuscript is now acceptable for publication, you may indicate that here to bypass the “Comments to the Author” section, enter your conflict of interest statement in the “Confidential to Editor” section, and submit your "Accept" recommendation.

Reviewer #2: All comments have been addressed

2. Is the manuscript technically sound, and do the data support the conclusions?

Reviewer #2: Yes

3. Has the statistical analysis been performed appropriately and rigorously? 

Reviewer #2: Yes

4. Have the authors made all data underlying the findings in their manuscript fully available?

Reviewer #2: Yes

5. Is the manuscript presented in an intelligible fashion and written in standard English?

Reviewer #2: Yes

6. Review Comments to the Author

Reviewer #2: Dear Authors, thank you very much for your review. All my comments have been addressed and I'm fully happy with this new version.

7. PLOS authors have the option to publish the peer review history of their article (what does this mean?). If published, this will include your full peer review and any attached files.

Reviewer #2: Yes: Giorgia della Rocca

---

## [Editor Report · Acceptance letter]

13 May 2020

PONE-D-19-30670R2 

Validation of the UNESP-Botucatu pig composite acute pain scale (UPAPS) 

Dear Dr. Luna:

I am pleased to inform you that your manuscript has been deemed suitable for publication in PLOS ONE. Congratulations! Your manuscript is now with our production department. 

With kind regards,

on behalf of

Dr. Francesco Staffieri 

Academic Editor

PLOS ONE